# Unseasonal atmospheric river drives anomalous summer snow accumulation on glaciers of the subtropical Andes

Claudio Bravo[1], Sebastián Cisternas[1], Maximiliano Viale[2], Pablo Paredes[1], Deniz Bozkurt[3,4,5], Nicolás García-Lee[1,6]

[1]Centro de Estudios Científicos, Valdivia, Chile.
[2]Instituto Argentino de Nivología, Glaciología y Ciencias Ambientales, Mendoza, Argentina.
[3]Departamento de Meteorología, Universidad de Valparaíso, Valparaíso, Chile.
[4]Centro de Ciencia del Clima y Resiliencia (CR)2, Santiago, Chile.
[5]Center for Oceanographic Research COPAS COASTAL, Universidad de Concepción, Concepción, Chile
[6]MSc. Program in Water Resources, Universidad Austral de Chile, Valdivia, Chile.

*Correspondence to*: Claudio Bravo (cbravo@cecs.cl)

**Abstract.** Climate change is associated with changes in the frequency and intensity of extreme weather events. Extreme weather is impacting the mass balance of Andean glaciers, a phenomenon that requires further detailed investigation. Among these extreme events, atmospheric rivers (ARs) play a significant role, potentially leading to either accumulation or melting events on glaciers. To assess the impact of ARs on Andean glaciers, we analysed an unseasonal event that occurred at the end of January 2021, marked by extreme snowfall in the highlands and heavy rainfall, landslides, and flash floods in the lowlands, during the typically dry austral summer period. Satellite imagery and meteorological observations in the glaciated Maipo River basin and its Olivares River sub-basin (33°S) enabled the characterisation of this event and its basin-scale impacts. Moreover, a glacier mass balance model allows us to quantify the effects of the AR on the Olivares Alfa Glacier (4284 to 4988 m a.s.l.) in the context of the preceding six hydrological years. The large water vapour transport by the AR led to substantial snow accumulation on the Maipo River glaciers, resulting in a post-event snowline observed at 2463 m a.s.l. In the Olivares River sub-basin, the 0°C isotherm dropped from typical summertime altitudes of 4000-4500 m.a.s.l. to 3250 m a.s.l. during the event; below the frontal zone of all glaciers in this sub-basin. The mass balance model for the Olivares Alfa Glacier during the dry 2020/21 hydrological year showed a trend toward negative values at the beginning of the ablation season, aligned with previous years and the prevailing severe drought conditions. However, the AR snowfall event combined with cooler conditions during the remainder of the ablation season compared to previous years, offset this trend and brought the mass balance closer to equilibrium. This demonstrates that an unseasonal snow accumulation event can significantly counteract the broader seasonal trends affecting subtropical Andean glaciers. Our study sheds light on the impacts of extreme and unseasonal snow accumulation events on glacier mass balance in the high Andes, particularly those associated with ARs, a synoptic feature projected to become more common in a warming climate.

# 1 Introduction

Glaciers, with their high sensitivity to climatic variations, stand as crucial indicators of climate change in regions spanning from tropical to polar areas. Central to their behaviour is the mass balance, a critical metric capturing the net gain or loss of ice and snow over specific periods. The mass balance, in turn, is profoundly influenced by the prevailing atmospheric conditions. Over the last two decades, observations in the Andes have shown a marked tendency to glacier mass loss, largely attributed to increases in air temperature and/or decreases in precipitation (Braun et al., 2019; Dussaillant et al., 2019). Embedded in this overall negative mass balance trend, there is a large interannual variability in atmospheric conditions, and so in the annual mass balance of glaciers.

In the subtropical Andes, between 32°S and 36°S in Chile and Argentina, the atmospheric interannual variability is mainly modulated by global-scale atmospheric-oceanic circulation patterns such as the El Niño Southern Oscillation (ENSO), the Southern Annular Mode (SAM), and the Interdecadal Pacific Oscillation (IPO), among others (Garreaud et al., 2009). For instance, the warm ENSO phases (El Niño) are usually associated with positive anomalies in snowfall in this region, while cold ENSO phases (La Niña) can result in below-average snow accumulation (Masiokas et al., 2020). Besides global-scale phenomena modulating annual mass balance, extreme synoptic-scale events, such as intense heat waves or precipitation events, can strongly modulate annual mass changes of Andean glaciers (Gonzalez-Reyes et al., 2024; Poveda et al., 2020). To our knowledge, the impact of individual extreme events on the annual mass balance of Andean glaciers has not been assessed yet, despite their increasing magnitude and frequency with climate change (Poveda et al., 2020).

While significant advances have been made in understanding extreme temperature and precipitation events in the subtropical Andes and their associated synoptic conditions (Bozkurt et al. 2016; Viale et al., 2018; Feron et al., 2019; Jacques-Coper et al., 2016 and 2021; Demortier et al., 2021; Valenzuela et al. 2022; Gonzalez-Reyes et al., 2023; Garreaud et al., 2024), their influence on glaciers mass balance remains largely unexplored. One pertinent subject of inquiry relates to the occurrence of atmospheric rivers (ARs) leading to intense precipitation events in the subtropical Andes (Viale et al., 2018). ARs are narrow, elongated corridors of intense water vapour transport, often situated ahead of ocean cold fronts which primarily make landfall and discharge their vapour as precipitation on the western coasts of the mid-latitude continents (Guan and Waliser, 2019). The most intense ARs typically lead to extremely heavy orographic precipitation on mountainous western coasts, and so to severe hydrometeorological hazards (Ralph et al., 2006; Leung and Qian, 2009; Neiman et al., 2011; Payne et al., 2020).

In the subtropical Andes, ARs constitute the main source of water, particularly during the winter season (Viale et al., 2018; Saavedra et al., 2020). Indeed, Saavedra et al. (2020) quantified that AR events account for about 50% of the total annual snow accumulation and are 2.5 times more intense snowfall events than non-AR snowfall events. However, their effects on mid-latitude glaciers are complex and can be opposing, as they can bring both significant snowfall accumulation and extreme ice/snow melting (Guan et al. 2010; Little et al., 2019; Kropač et al., 2021).

The role of ARs in shaping glacier mass balance is complex and contingent upon multiple factors including synoptic conditions, seasonal time occurrences of the AR events, and the altitudinal and latitudinal location of the glaciers. Little et al. (2019)

showed that the days with the most intense ablation and largest accumulation rates at Brewster Glacier coincide with ARs that transport exceptionally high levels of water vapour, depending on their associated thermal conditions. In the case of ablation, water vapour and associated high temperatures transported poleward by ARs supply energy for melting. ARs can lead to glacier and snow melting through several mechanisms, including large-scale sensible heat transport (e.g., Bozkurt et al., 2022; Wille et al., 2022), rain-on-snow events (through rain heat flux increase) and an increase in cloud cover and downward longwave

radiation over mountains (Chen et al., 2019; Zou et al., 2021; Kropač et al., 2021; Wille et al., 2022). Conversely, when favourable conditions exist such as glaciers located above the 0°C isotherm, ARs can trigger substantial snow accumulation events.

       This dual role of ARs in driving both high accumulation and melting events is not limited to mid-latitudes. AR events similarly exert a two-fold influence in polar latitudes, such as Antarctica and Greenland. These events can contribute to increased

snowfall accumulation on the vast ice sheets, potentially mitigating their mass loss (Mattingly et al., 2018; Wille et al., 2021; Adusumilli et al., 2021; Maclennan et al., 2022). Conversely, ARs have been implicated in driving episodes of extreme warm temperatures, intense surface melting and liquid precipitation events in West Antarctica, the Antarctic Peninsula, and Greenland, particularly along the coastal zones, causing concerns about ice sheet stability (Wille et al., 2019, 2022; Xu et al., 2021; Box et al., 2022; Mattingly et al., 2023). Furthermore, ARs can initiate post-event feedback mechanisms on glacier

surfaces, for example, notably contributing to glacier albedo changes (e.g., Box et al., 2022). Analysing and quantifying these processes are essential for understanding glacier responses to extreme weather events.

       Beyond the scientific realm, understanding the impact and role of ARs on the Andean cryosphere holds profound societal significance. In the sub-tropical Andes, glaciers constitute natural water reservoirs, storing vast amounts of freshwater as ice, which gradually feeds rivers, supplying water for agriculture, drinking, and hydropower generation for over 10 million people

(Ayala et al., 2020; Crespo et al. 2020). Therefore, ARs can influence glacier mass balance, impacting the timing and magnitude of freshwater release for managing this vital resource. Furthermore, ARs can lead to extreme events such as floods and landslides triggered by rain-on-snow events (e.g., Guan et al., 2016; Dolant et al., 2017; Somos-Valenzuela et al., 2020; Rutllant et al., 2023), with devastating consequences for communities, infrastructure, and economies. Assessing these AR impacts contributes to a deeper understanding of glacier hazards triggered by extreme weather events (e.g., Carey et al., 2012)

and can help to enhance the long-term sustainability and resilience of regions reliant on glacier meltwater in the subtropical Andes.

       In the subtropics, ARs are much more frequent in winter, while precipitation and AR frequency are almost absent in summertime (dry season) over the western slopes of the subtropical Andes and the central Chilean lowlands (Viale and Garreaud, 2014; Viale et al., 2018). Despite these overall characteristics, episodes of intense precipitation can occasionally

occur in summer (Poveda et al., 2020). In late January 2021, the subtropical Andes of Chile and Argentina experienced a major and unusual precipitation event triggered by an unseasonal and zonal-oriented AR (Valenzuela et al., 2022). During the event, precipitation totals in central-southern Chile exceeded 100 mm over four days (January 28-31) in central-southern Chile (Valenzuela et al., 2022). The storm's first phase (28–29 January) occurred with a winter-like frontal system and its associated

AR but in a relatively warm, summertime environment; while during the second phase (30–31 January), the AR quickly dissipated, leaving residual moisture and forming a cut-off low, that fueled intense convection, rain, lightning, and hail, leading to significant societal disruptions and economic losses (Valenzuela et al., 2022). This extraordinary AR-related event represents a rare climatic anomaly, making the storm unusually strong even by winter standards, with historical records documenting similar storms occurring only 2–3 times in the past century (Valenzuela et al., 2022). This AR serves as the focal point of our study, as we assess its influence on the annual mass balance of the Olivares Alfa Glacier.

In this work, our objective is to evaluate the role of this extraordinary dry-season precipitation event, triggered by an AR, over the glaciers of the Maipo River basin, especially over the Olivares Alfa Glacier located in the Andes at 33°S (Fig.1). First, we describe the extraordinary nature of the event and the unusual large precipitation rate for summertime in the Andes at this latitude. We assess whether accumulation or melting during this event predominantly influences the annual mass balance. Secondly, we analyse the drivers behind the glacier's mass balance response to the AR event by quantifying the surface energy balance. Thirdly, we quantify how a single extreme unseasonal precipitation event can alter the mass balance trend for this particular year and in the context of seven hydrological years. Finally, we discuss the implications of this event, particularly in the context of the ENSO variability mode, a large-scale glacier mass balance forcing in this region. To accomplish these research objectives, we use remote sensing, radiosonde data, reanalysis data and available meteorological observations located in the glacierised high Olivares River sub-basin (Fig. 1). These meteorological data were also used as input for the COSIPY v1.3 mass balance model (Sauter et al., 2020), which allowed us to quantify the impact of this AR-related precipitation on the annual mass balance of the Olivares Alfa Glacier. To our knowledge, this study represents the first comprehensive examination of AR impacts on glacier mass balance in the Andean region.

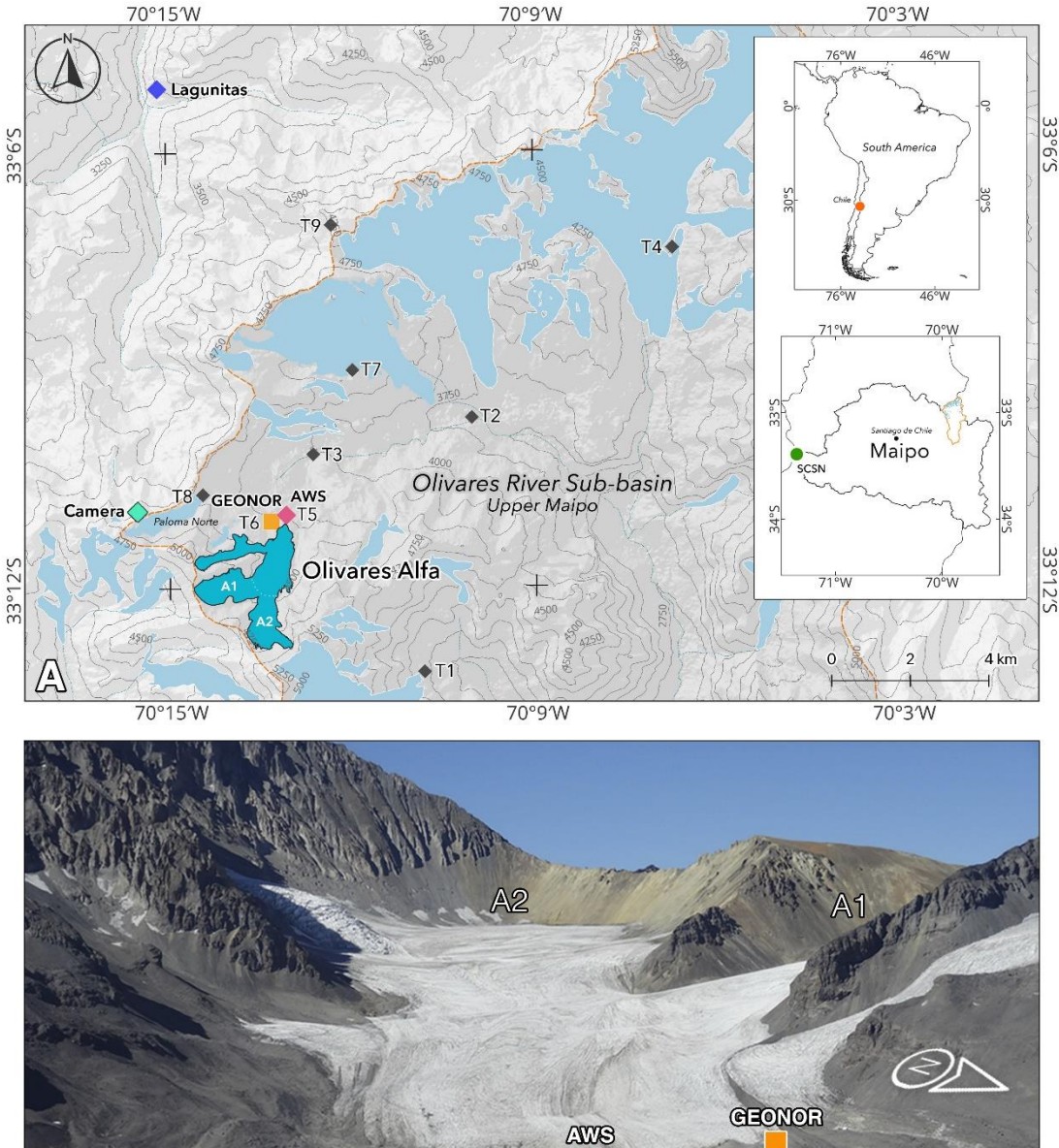


**Figure 1. A) Glaciers (light-blue colour) of the Olivares River sub-basin and the location of the Automatic Weather Station (AWS, magenta colour) and pluviometre (GEONOR, orange colour) close to Olivares Alfa Glacier (in dark blue). T1 to T9 mark the locations of the air temperature sensors (T5 is the AWS), and A1 and A2 are the two cirques that form the accumulation zone of the Olivares-Alfa glacier. The purple diamond at the top left indicates the location of Lagunitas weather station, and the green diamond**

**is the location of a camera over Paloma Norte Glacier. The upper inset provides the regional context of the Maipo River basin in South America, and the bottom inset highlights the Maipo Basin (black line) and the Olivares River sub-basin (orange line). SCSN (Santo Domingo) corresponds to the site from which radiosondes are launched. B) is a photograph taken on 5 February 2015 by the author with a detailed view of the Olivares Alfa Glacier.**

**2 Study area**

We conducted our assessment at two different river basin scales: first at the Maipo River basin and then at the Olivares River sub-basin of the Maipo River (Fig. 1). The Maipo River is the main water source for Santiago, Chile's densely populated capital, supporting its residents, agriculture, and industry. Of the total of the Maipo River basin water, around 60% is used in the agriculture sector and 35% is for drinking and sanitation (Alvarez-Garreton et al., 2024). The basin contains around 1000 ice bodies comprising a total glacier area of 388 km$^2$ (Barcaza et al., 2017). Our analyses focused on the Olivares River sub-

basin, which hosts a glacierised area of approximately 70 km$^2$ centred around 33°10' S 70°10'W. Within the Olivares River sub-basin lies the highest glaciers in the Maipo River basin, ranging from 3500 to 5800 m a.s.l. (Ayala et al., 2020), whose ice masses have experienced the largest ice loss compared to glaciers in other Maipo River sub-basins between 2000 and 2013 (Farías-Barahona et al., 2020).

The Olivares River sub-basin is home to several glaciers, including Olivares Alfa, Olivares Beta, Olivares Gamma, and Juncal

Sur (Fig. 1). These glaciers exhibit clear signs of thinning and retreat (Farias-Barahona et al., 2020; Malmros et al., 2016). Additionally, they have shown a trend towards surface darkening since the 1980s (Shaw et al., 2021; Barandun et al., 2022). Glacier simulations, detailed in the next section, focused on the Olivares Alfa Glacier (Fig.1), a mountain glacier with the accumulation zone divided in two cirques, both primarily oriented towards the northeast and with a mean elevation of 4574 m a.s.l. According to Shaw et al. (2021), the estimated area of this glacier as of March 2020 is 3.2 km$^2$. Satellite images and aerial

photographs reveal that the Olivares Alfa was formerly part of a continuous glacier that extended over most of the western headwater of the sub-basin. Malmros et al. (2016) quantified a significant glacier area loss of 63% between 1955 and 2013 and indicating a gradual fragmentation of the ice mass over the years.

**3 Materials and methods**

**3.1 Catalogue of Atmospheric River events and extraordinary total summer precipitation for one event**

To contextualize the extraordinary occurrence of the summer-2021 AR, we use ERA5 reanalysis data (Hersbach et al 2020) to catalogue the ARs in summer and demonstrate the climatological low frequency of this synoptic feature. Further, we analysed how the precipitation total during this event was also extraordinary for summertime in the Andes at this latitude.

For the first part, we identified AR conditions on the Pacific coastal grid points of the ERA5 reanalysis data (Hersbach et al 2020), in central Chile (30°S-35°S) over the 1941-2023 period (83 years). AR conditions are classified into five categories

according to the scale proposed by Ralph et al. (2019). The scale combines the maximum moisture transport integrated over the vertical column of the atmosphere, i.e., the Integrated Vapour Transport (IVT), with the duration of the ARs conditions, determined as the period during which IVT remains consistently greater than 250 kg m$^{-1}$ s$^{-1}$. Category 1 are primarily associated with beneficial precipitation-related impacts, while Category 3 can be a balance between beneficial and hazardous and Category 5 are primarily hazardous. For further details on the ARs scale see Ralph et al (2019). In our study, we retain ARs

events if at least Category 1 conditions occurred at two or more of the coastal grid points between 30° and 35°S. AR conditions in central Chile typically affect multiple grid points within this latitude range. Therefore, the onset and demise of each AR event is established by the start and end times of the overlapping AR conditions across all relevant coastal grid points. For further analyses, we use two variables to measure AR event intensity: the maximum AR category reached during the event and the maximum instantaneous IVT value among all grid points with AR conditions. We focused on evaluating the summer

AR events over the available 83-year period. Further, we analysed the impact of past ARs on the simulated glacier mass balance in each hydrological year between 2014/15-2020/21 across all the seasons. This AR catalogue is shown in Table S1. To remark on the extraordinary precipitation during summer in the Andes, we used available data from the Lagunitas weather station (33.08° S, 70.25° W, see Fig. 1) for the period 1959-2023. We defined a precipitation event when the daily precipitation rate is equal to or larger than 2 mm for one or more consecutive days. Although this station is located at a lower elevation

(2765 m a.s.l.), it provides one of the longest precipitation records in this region of the Andes, allowing us to place the summer-2021 event in the context of an observed precipitation climatology.

### 3.2 Basin-scale analysis

Firstly, at the Maipo River basin scale (Fig. 1a), we aim to estimate if glaciers were subject to accumulation and/or melting. For this, we utilised the MODIS daily snow cover product (Terra MOD10A1 and Aqua MYD10A1, V0006, 500 m resolution)

(Hall et al., 2006) to derive the snowline altitude in the Maipo River basin. The snowline altitude was calculated using the method developed by Krajčí et al. (2014) over the period from April 2001 to March 2021. MODIS snow product it is known to be susceptible to cloud cover, limiting its accuracy in areas with frequent cloud coverage. However, the approach proposed by Krajčí et al. (2014) overcomes this limitation by offering improved tolerance to clouds. By allowing a higher cloud cover threshold (90% in our case) this method can still provide reliable snow cover information, even in cloud-prone regions. This

enhanced cloud tolerance results in better accuracy compared to other snow detection methods, such as those by Parajka et al. (2010) and Da Ronco and De Michele (2014), which perform less effectively under heavy cloud cover. This approach enables us to assess the exceptional nature of the AR event at a regional scale and in the context of the past 20 years.

Secondly, we estimated the altitude of the 0°C isotherm or freezing level to assess the glacier area experiencing melting by using radiosonde and in-situ data. The radiosonde data registered by the Santo Domingo station (33.63°S, 71.65°W, Fig. 1a)

were obtained from the NOAA Integrated Global Radiosonde Archive (IGRA) database. We utilised air temperature data from an Automatic Weather Station near the front of the Olivares Alfa Glacier (AWSOA, 4220 m a.s.l., 33°10' S, 70°13' W, Fig. 1), as well as multiple air temperature sensors located at different altitudes in the Olivares River sub-basin (3606, 3663, 4004, 4020, 4240, 4288, 4453, 4466 and 4772 m a.s.l., T1 to T9 on Fig. 1a) to estimate the freezing level. The freezing level was estimated using a linear regression of the hourly air temperature records at different altitudes. To ensure accuracy and

reliability, we compared the 0°C isotherm estimated from the Santo Domingo radiosonde data with the one derived from the air temperature sensors in the sub-basin.

The estimated snowline altitude and freezing level enable us to estimate the glacier area experiencing melt and/or accumulation before, during, and after the precipitation event in the Maipo River basin and the Olivares River sub-basin. The estimated snowline and freezing level were then compared with the hypsometry of the basin's glaciers, derived from glacier outlines from the year 2021 and the AW3D30 digital elevation model (DEM) data from the Japan Aerospace Exploration Agency.

### 3.3 Glacier-scale analysis

To assess the impact of the precipitation event on the glacier mass balance, we utilised the coupled snowpack and ice surface energy and mass balance model "COSIPY" (v1.3), developed by Sauter et al. (2020). This model employs state-of-the-art equations and parameterizations to estimate energy and mass balance. Input data for the model were collected from two sources near the Olivares Alfa Glacier: the AWSOA and the precipitation sensor GEONOR T200B-M (see Table 1 for sensor details and Figure 1 for location). These observations include air temperature, air relative humidity, wind speed, atmospheric pressure, precipitation, incoming shortwave and longwave radiation (Table 1, Fig. S1). The model distributes the meteorological variables over the glacier surface at a spatial resolution of 100 m. Direct longwave radiation measurements became available starting in September 2016. Therefore, data derived from ERA5 reanalysis were utilised after undergoing bias correction with the aid of direct observations (Lopes et al. 2022; Fig. S1g) to cover the complete period from April 2014 to March 2021. Consequently, we simulated the total mass balance of seven hydrological years.

Furthermore, glacier outlines were obtained from medium and high-resolution imagery for the modelling period of 2013/14 to 2020/21 (see Table S2), along with the AW3D30 DEM. Due to the scarcity of detailed and consistent snow-depth observations on the glacier surface at the beginning of each hydrological year (i.e., April in the subtropical Andes), we adopted annual time-steps when running the model. This entailed initialising the model with a free-snow starting condition for each year of the period of interest.

The analysed model outputs include glacier total mass balance (in m w.e.) and surface energy fluxes, such as turbulent fluxes and net short and longwave radiation (in W m$^{-2}$). These model outputs were analysed to assess the impact of the event on the energy balance and on the glacier's annual mass balance. In addition, a comparison of the annual mass balance series was made with that of previous hydrological years. Total mass balance was assessed rather than surface mass balance, to incorporate feedback and changes in internal mass balance, melt and refreezing processes (Sauter et al., 2020). To further evaluate the impact of this unseasonal precipitation event, we also estimated the 2020/21 hydrological year's mass balance under a hypothetical scenario without the AR's influence, removing the accumulation attributed to the summer-2021 AR. To complete the last two months of the 2020/21 hydrological year, the mass balance time series from previous similar years (i.e., those years with negative mass balance until the end of January, 2017/18, 2014/15 and 2019/20) were decomposed to extract the trend for each year (Box et al., 2015). Then, the 2020/21 mass balance series was detrended, and the average, maximum, and minimum trends derived from previous years, were applied to the analysed hydrological year to hypothesize a scenario range without the occurrence of the AR.

**Table 1. Sensors of the Automatic Weather Station located near the front of Olivares Alfa Glacier (AWSOA), at 33°10' S, 70°13' W, 4220 m a.s.l.**

| Variable | Units | Instrument Type | Manufacturer | Model | Accuracy |
|---|---|---|---|---|---|
| Air temperature (AirT) | °C | Temperature and relative humidity probe | Vaisala | HMP 155 | ±0.2 °C |
| Relative humidity (RH) | % | Temperature and relative humidity probe | Vaisala | HMP 155 | ±2% |
| Wind speed (WS) | m s$^{-1}$ | Wind monitor | R. M. Young | Heavy Duty Wind Monitor-HD-Alpine 05108-45 | ± 0.3 m/s or 1% of reading |
| Wind direction (WD) | degrees | Wind monitor | R. M. Young | Heavy Duty Wind Monitor-HD-Alpine 05108-45 | ± 3 degrees |
| Precipitation (PP) | mm | Precipitation - rain gauge | Geonor | T200B-M | ±0.1% |
| Incoming Shortwave Radiation (SWR) | W m$^{-2}$ | Pyranometer | Kipp and Zonen | CMP3 | ±10% |
| Incoming Longwave Radiation (LWR) | W m$^{-2}$ | Pyrgeometer | Kipp and Zonen | CGR3 | ±10% |
| Atmospheric pressure (AP) | hPa | Barometer | Vaisala | PTB-110 | ±0.3 hPa at +20 °C |

## 4 Results

### 4.1 Unseasonal and zonal Atmospheric River event and extraordinary accumulated precipitation

A climatological analysis of the AR conditions on the Pacific coastal grid points (30°S-35°S, Fig 2a) using ERA5 reanalysis data reveals the low frequency of summer AR events in central Chile. Between 1941 and 2023 (83 years), only 19 weak AR events occurred during the summer season (DJF), corresponding to AR categories 1 or 2 based on Ralph et al. (2019). This frequency roughly translates to one AR event every four summers. In contrast, the total AR events during all seasons reached 687, highlighting the dominance of AR activity outside of summertime in central Chile.

The AR event of January 2021, however, stands out due to its extreme characteristics, making landfall in central Chile during the middle of the austral summer. According to the long precipitation time series at the nearby Lagunitas weather station (see Fig.1a for location), a total precipitation of 95 mm was recorded, marking it as the second largest summertime precipitation event since 1959, with a return period of 30 years or more (Fig. 2b). Valenzuela et al. (2022) also showed the extreme values

of this precipitation event in highlands and lowlands stations (see their figure 7). Three of the top four summertime precipitation events at Lagunitas station since 1959 were caused by unseasonal ARs (Fig. 2b). These ARs were among the largest transporters of cross-barrier water vapour flux to the Andes (Fig. 2a), indicating the key role of orographic lifting in enhancing precipitation intensity in the Andes.

The synoptic characteristics of the 2021 event are shown in Figure 3a. Sea level pressure and IVT at 1200 UTC on 29 January 2021 show the AR's zonal orientation, which was nearly perpendicular to the Andes. Such alignment is crucial for efficiently lifting moist air over the mountains, leading to intense precipitation (Garreaud et al., 2024). According to the ERA5-based AR summertime catalogue, the AR of January 2021 was the most zonal-oriented of the 19 summertime AR events in the last 83 years (Fig. 3b). Additionally, the third most zonally oriented AR was the AR of 1965 which is the number one precipitation
event in this 83-year period (Fig. 3b and Fig. 2b), denoting the high importance of the AR orientation to augment orographic effects on precipitation. The 2021 AR was classified as Category 1 along the coastal grid points between 34.5°S and 39°S during its peak (Fig. 3b).

The storm's cumulative precipitation over 120 hours (27 January 0000 UTC to 01 February 0000 UTC) is shown in Figure 3b. Precipitation totals exceeded 100 mm in some regions of the central Andes, with widespread impacts across both mountain
and lowland areas. The orographic interaction with the AR's zonal vapour transport (Fig. 3a) highlights the significant enhancement of precipitation due to the Andes' barrier effect.

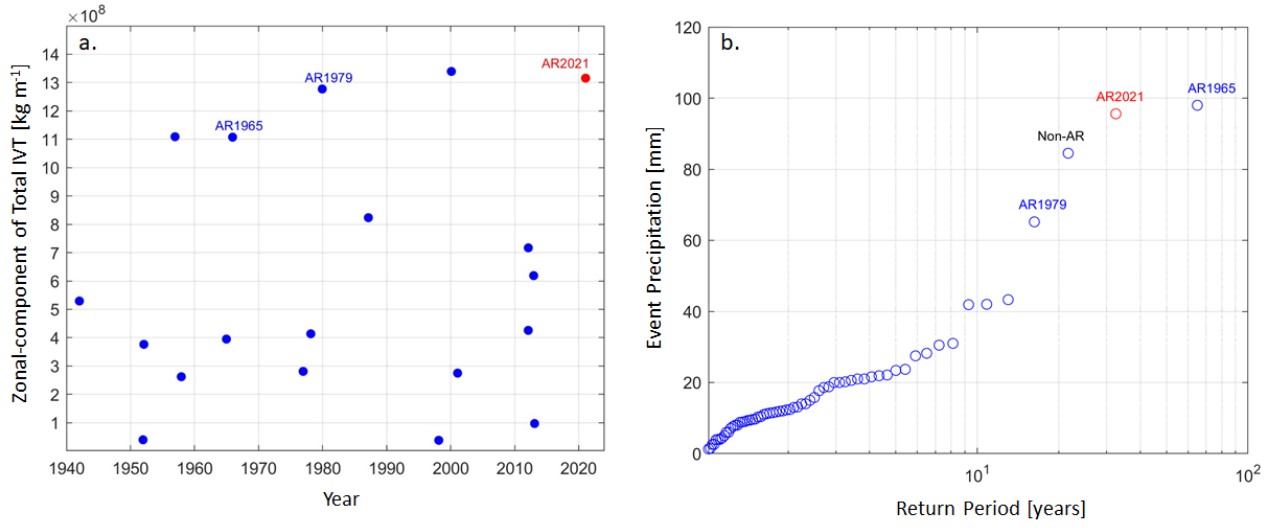

**Figure 2. a) Cross-barrier (zonal component) of the Total Integrated Vapour Transport during the 19 summertime (DJF) AR events**
**occurred over the 1941-2023 period (83 years). b) Return period of summer (DJF) precipitation events (mm) at Lagunitas station**
**(2765 m a.s.l., see Fig. 1 for location). Highlighted are the top four events, including the AR of this study (AR2021).**

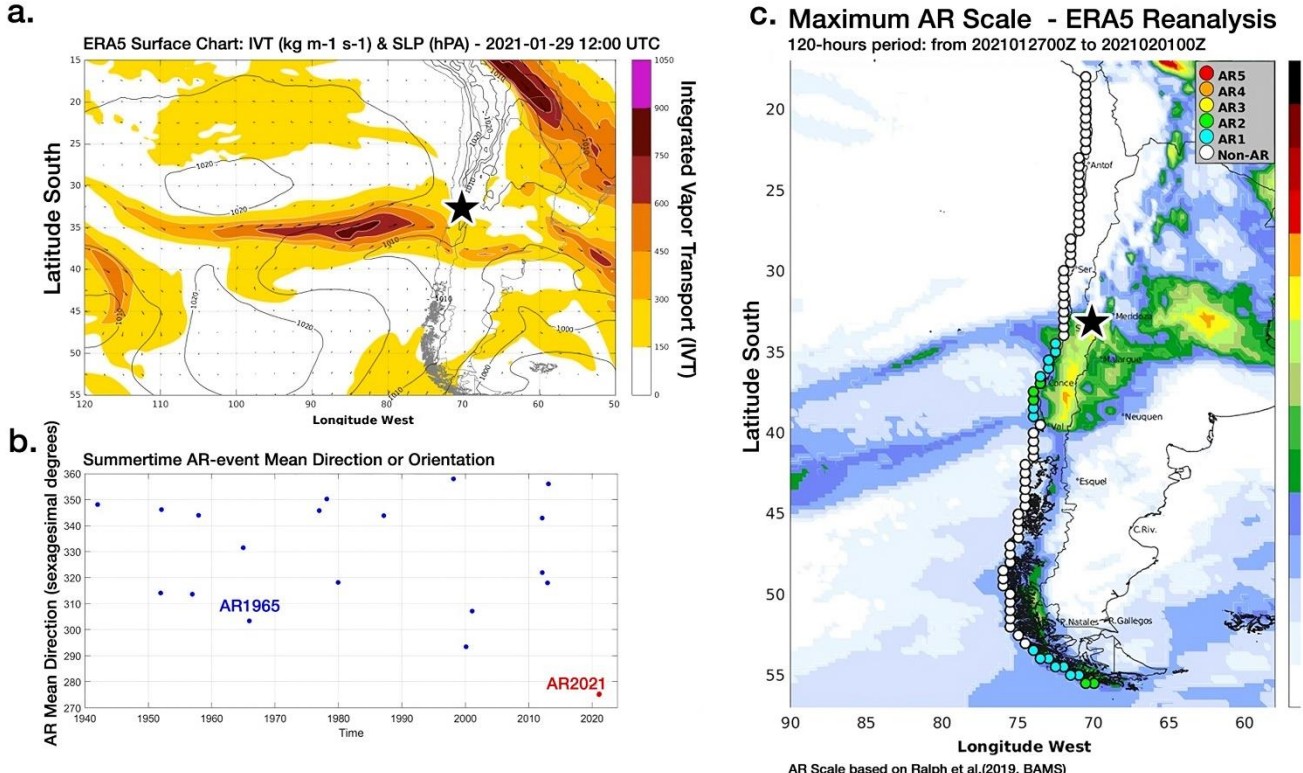

**Figure 3. ERA5 reanalysis shows the unseasonal AR that made landfall in central Chile. a) Surface chart with the Sea Level Pressure (SLP, lines in hPa) and the Integrated Vapour Transport (IVT, shaded in kg m⁻¹s⁻¹) at 1200UTC on 29 January 2021. b) Mean event direction during the 19 AR summertime (DJF) events that occurred over the 1941-2023 period (83 years). Direction expressed in sexagesimal degrees, being 360° from the North to South and 270° from the West to East. c) Storm total precipitation accumulated from 0000 UTC on 27 January to 0000 UTC on 01 February 2021 (shaded in mm). The colour-coded circles on the Pacific coast show the AR scale (Ralph et al. 2019) reached by the AR. Black star indicates the location of the Olivares River basin.**

## 4.2 Snowline altitude in the Maipo River basin estimated from MODIS satellite data

In the Maipo River basin, the snowline altitude  from MODIS data was estimated to be around 4700 m a.s.l. before the summer-2021 AR event (Fig. 4). This altitude  is over the 75th percentile of the distribution of the snowline for the January and February months between 2001 and 2021, where the snowline typically ranges between 3900 and 4500 m a.s.l., with higher glacier sections usually being snow-covered during these months. The day previous to the event the snowline was 4693 m a.s.l. and the mean of the 10 days previous to the event was 4654 m a.s.l. During the event, the snowline altitude sharply decreased, reaching a minimum altitude of 2463 m a.s.l., which is lower than the 0°C isotherm minimum (~3250 m a.s.l., see next section). Immediate post-event measurements placed this value among the lowest January and February snowline altitudes observed from MODIS SNOW for the period 2001 to 2021 (Fig. 4). The average summer snowline altitude  for this period is 4250 m a.s.l. After the event, the snowline altitude gradually increased reaching an altitude similar to January 2021 (~4200-4400 m a.s.l. with 4493 m a.s.l. as maximum value), but did not return to its pre-event altitude by the end of the hydrological year.

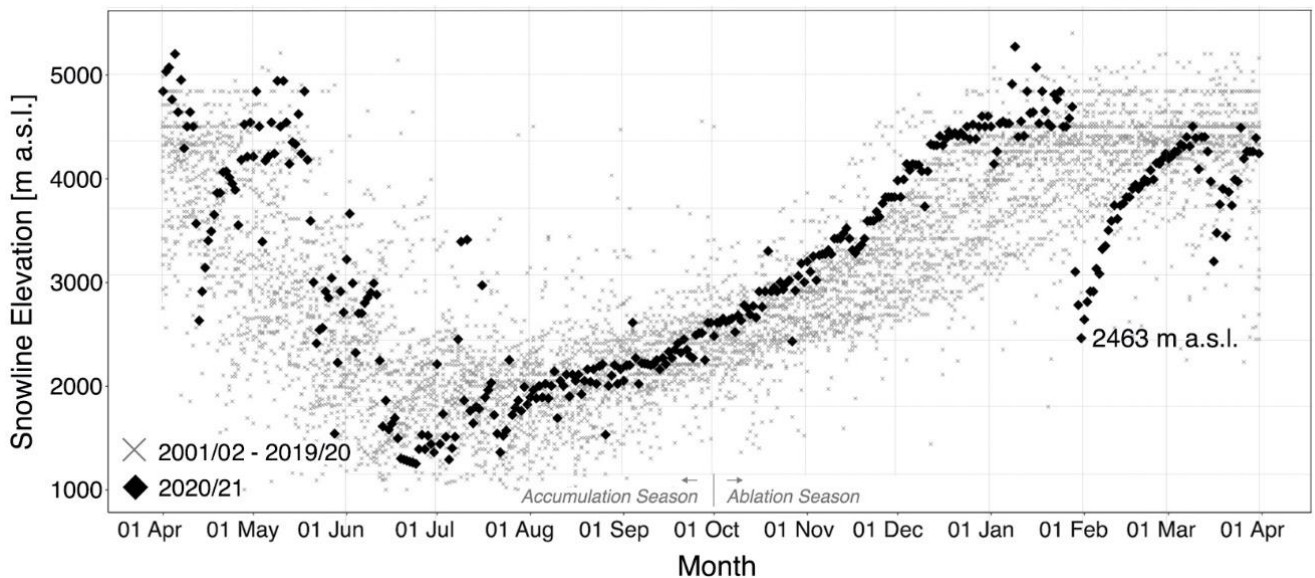

**Figure 4. Snowline time series at the Maipo River basin for the hydrological years between 2001 and 2019 (grey crosses) and 2020-2021 (black diamonds). The lower snowline altitude reaching during the event is indicated (2463 m a.s.l.).**

### 4.3 0°C isotherm altitude at the Olivares River sub-basin

In the days before the event (25 to 28 January), the 0°C isotherm altitude was estimated to be between 4000 and 4500 m a.s.l. with a marked diurnal cycle (Fig. 5). The 0°C isotherm altitude estimations from the in-situ air temperature sensors and radiosonde data were practically the same. At the onset of the event (i.e., 29 January), the 0°C isotherm decreased to around 3700-3900 m a.s.l., and the next day, it further decreased to 3250 m a.s.l. This last value corresponds to the minimum altitude of the 0°C isotherm during the event and coincides with the maximum precipitation rate recorded by the GEONOR sensor (Fig. 5). Around this time, all glacier areas in the Olivares River sub-basin were accumulating snow during the first half of the 30 January. A rapid increase in the 0°C isotherm altitude occurred after mid-day of 30 January, resulting in positive temperatures at the glacier lowest elevations, particularly in the frontal sections of the Olivares Gama and Juncal Sur glaciers, which extend to around 3800 m a.s.l. On 31 January, the 0° isotherm decreased to similar altitudes to those estimated in the previous day, oscillating between 3300-3500 m a.s.l., coinciding with the event's second-highest precipitation rate. Consequently, the total glacier area of the basin experienced negative air temperatures. On the days after 1 February, no precipitation was recorded, and the 0°C isotherm oscillated between 3500 to 3800 m a.s.l., until 3 February when it quickly rose to pre-event levels. During the post-event days (after 1 February), the radiosonde-derived 0°C isotherm was lower than the 0°C isotherm estimate from in-situ temperature sensors. This discrepancy could be related to the trajectories. According to HYSPLIT model (see Fig. S2), the 29 and 30 of January show that the radiosounding launched from Santo Domingo was in

the direction to the Andes, while the 2 of February shows a trajectory to the north-west, to the Pacific Ocean. We hypothesise that this difference in trajectories determines the discrepancy.

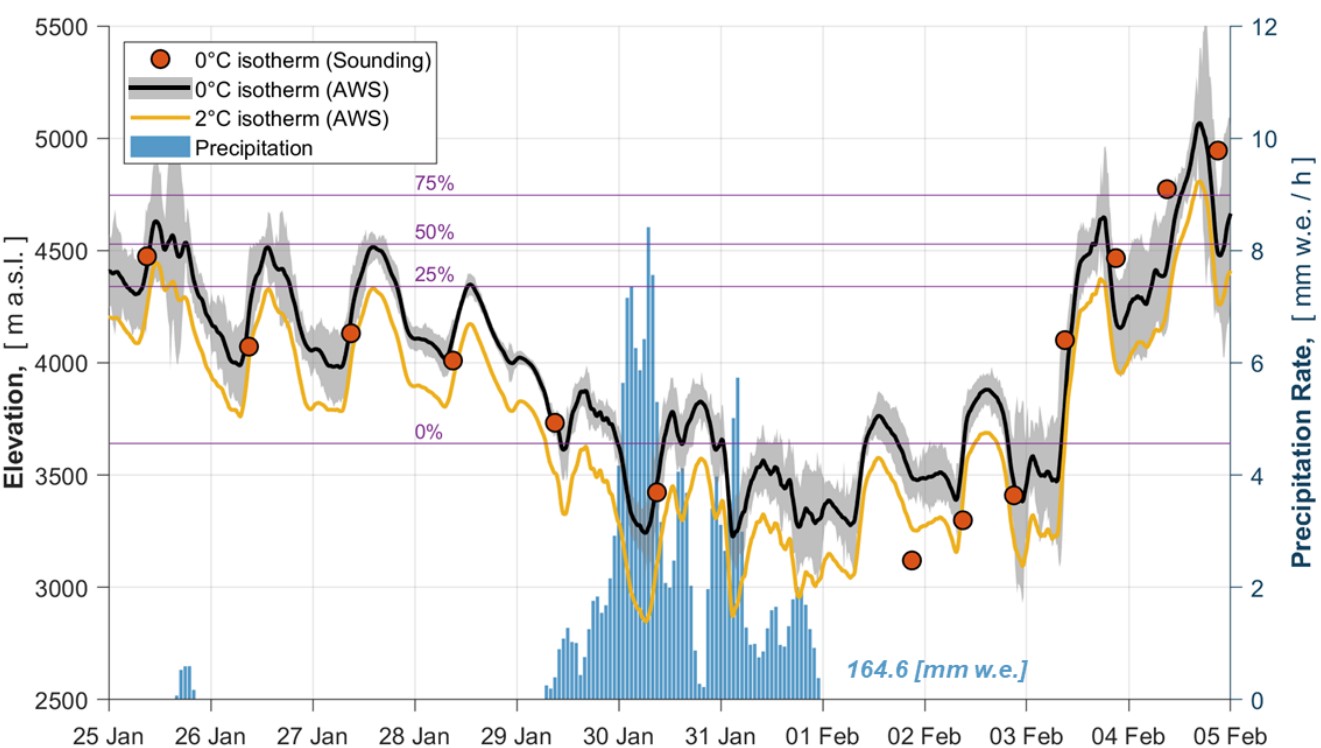

Figure 5. Time series of the 0°C isotherm (left axis) around the event. The black line corresponds to the calculated values using several air temperature sensors installed in the basin (Fig. 1). Red circles correspond to the 0°C isotherm obtained from the Santo Domingo radio sounding. Bars indicate the hourly precipitation rate (right axis) recorded by the GEONOR T200B-M. Percentages correspond to the glacier area hypsometry of the Olivares River sub-basin. As a reference the 2°C isotherm is also shown with a yellow line.

## 4.4 Surface energy flux changes during the event at the Olivares Alfa Glacier surface

Summertime (DJF) averaged modelled fluxes from 2014 to 2021 shown in Table 2 indicate that the main source of energy on the glacier surface is the net shortwave radiation (mean of 120 W m$^{-2}$), and the main energy loss is from the net longwave radiation (-53 W m$^{-2}$). Latent heat fluxes are predominantly negative (mean of -39 W m$^{-2}$), indicating that the glacier surface sublimates during the ablation season. Conversely, sensible heat flux is positive (mean of 18 W m$^{-2}$), reflecting a lower glacier surface temperature compared to the surrounding air temperature. Ground heat fluxes vary between positive and negative values during night time and daytime, respectively (Fig. 6). The long-term available melt energy typically reaches a mean value of 54 W m$^{-2}$ at this time of the year, with the maximum around 150 W m$^{-2}$, for instance, as estimated just before the AR event (Fig. 6).

During the summer-2021 AR event, these typical summertime values changed (Fig. 6, Table 2). There was a notable decrease
in both the net shortwave and longwave radiation on 29 and 30 January, coinciding with the maximum precipitation rate and
AR moisture transport (Figs. 2a and 5). The event's mean net shortwave and longwave radiation values during the event were
22 W m$^{-2}$ and -17 W m$^{-2}$, respectively. Continuous cloud cover during the storm reduced the variability in net longwave
radiation, which turned positive for a short period. Turbulent fluxes tended to decrease as well. Latent heat flux dropped (i.e.
became less negative) due to added atmospheric water vapour by the AR and the decrease of the melting rate in the glacier
surface, diminishing the humidity gradient between the surface and air. Similarly, sensible heat flux decreased because of the
reduced air temperature during the storm and the persistent cloud cover, lessening the temperature gradient between the surface
and air. During 30 and 31 January, negative values of sensible heat flux were estimated, indicating that the glacier surface was
warmer than the surrounding air. Melt energy was again available three days after the event.

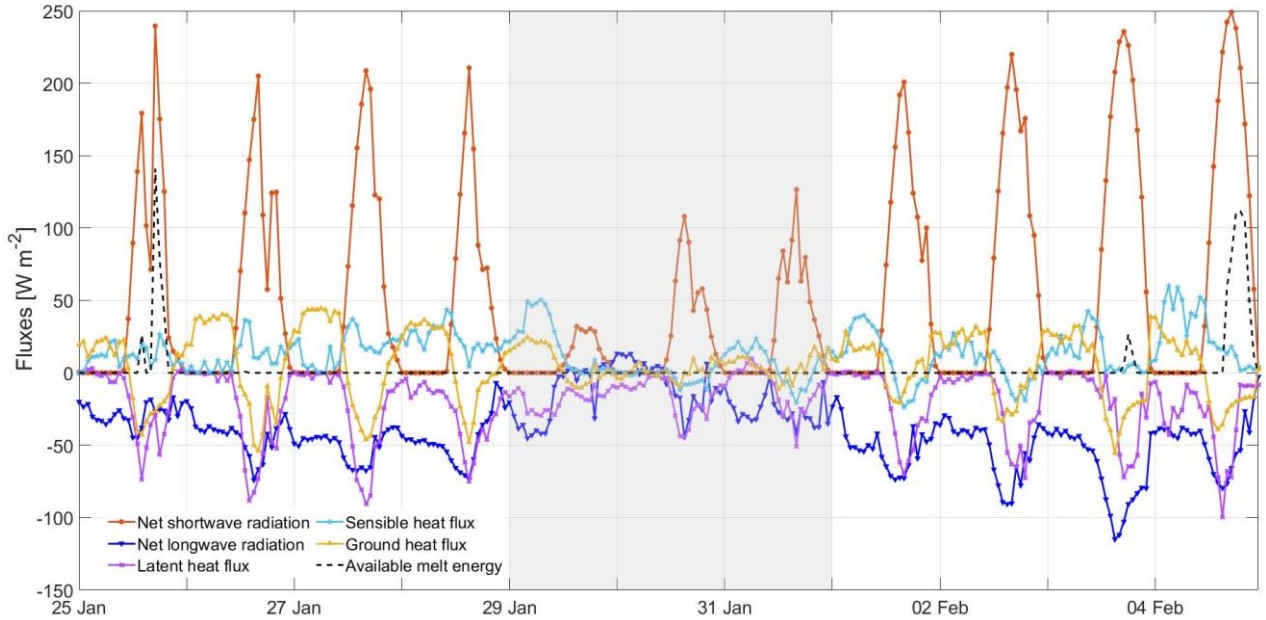

**Figure 6. Glacier energy balance surface fluxes estimated by COSIPY model forced with the Olivares Alfa AWS (Fig. 1). The grey
area corresponds to the dates of the AR event (Valenzuela et al., 2022).**

### 4.5 Simulation of the Olivares Alfa Glacier mass balance

The simulated mass balance of the Olivares Alfa Glacier for the hydrological years (April-March) between 2014/15 and
2020/21 are shown in Figure 7. Five of all seven years finished with negative mass balance (Fig. 7a), with the mass balance of
2019/20 year being the most negative year (-3.2 m w.e.). The mass balance in 2018/19 was barely positive, while the mass
balance in 2016/17 finished with a positive mass balance of 0.25 m w.e., with the mean of the 7-year period being -0.8 m w.e.
Typically, ablation in April is larger than accumulation (Fig 7c), occasionally interrupted by snow accumulation events

depending on the hydrological year (Fig. 7b). The 2015/16 year shows that the ablation ceased due to temperature drops, stabilising at around -0.5 m w.e. before the first accumulation event in July. Accumulation in the hydrological year 2016/17

was mostly due to a significant event in April.

The hydrological year 2020/21 shows a period of accumulation concentrated mainly during June (see Fig. 7b). Before this month, no important accumulation events were observed at the start of the hydrological year (April-May) dominated by ablation. After July, no important accumulation events were observed, keeping the mass balance close to equilibrium. In this hydrological year, the ablation started in September 2020, earlier than expected, albeit gradually. No accumulation events were

observed in the ablation season until the January 2021 AR event. In spring and part of the summer, the mass balance experienced a gradual decrease, primarily due to melting (Fig. 7c). This tendency was abruptly interrupted by the January 2021 AR event. Compared to previous years and due to the AR event, the mass balance of this hydrological year was among the least negative within the analysed period, like the 2015/16 and the 2018/19 hydrological years. The spatial distribution of the mass balance is shown in Figure S3. As expected, a spatial gradient in the mass balance is evident, with a consistently larger

mass loss in the frontal zone during all the years. The mass balance in 2014/15 and 2019/20 experienced the most significant surface mass loss.

We performed a hypothetical experiment in the mass balance simulation during the 2020/21 hydrological year, assuming the AR event did not occur (Fig. 8). In this hypothetical scenario, the mass balance would have ranged between -2.4 and -0.6 m w.e. (mean of -1.5 m w.e.), instead of the near zero observed, underscoring the significant impact of this singular accumulation

event in the ablation season on the annual mass balance.

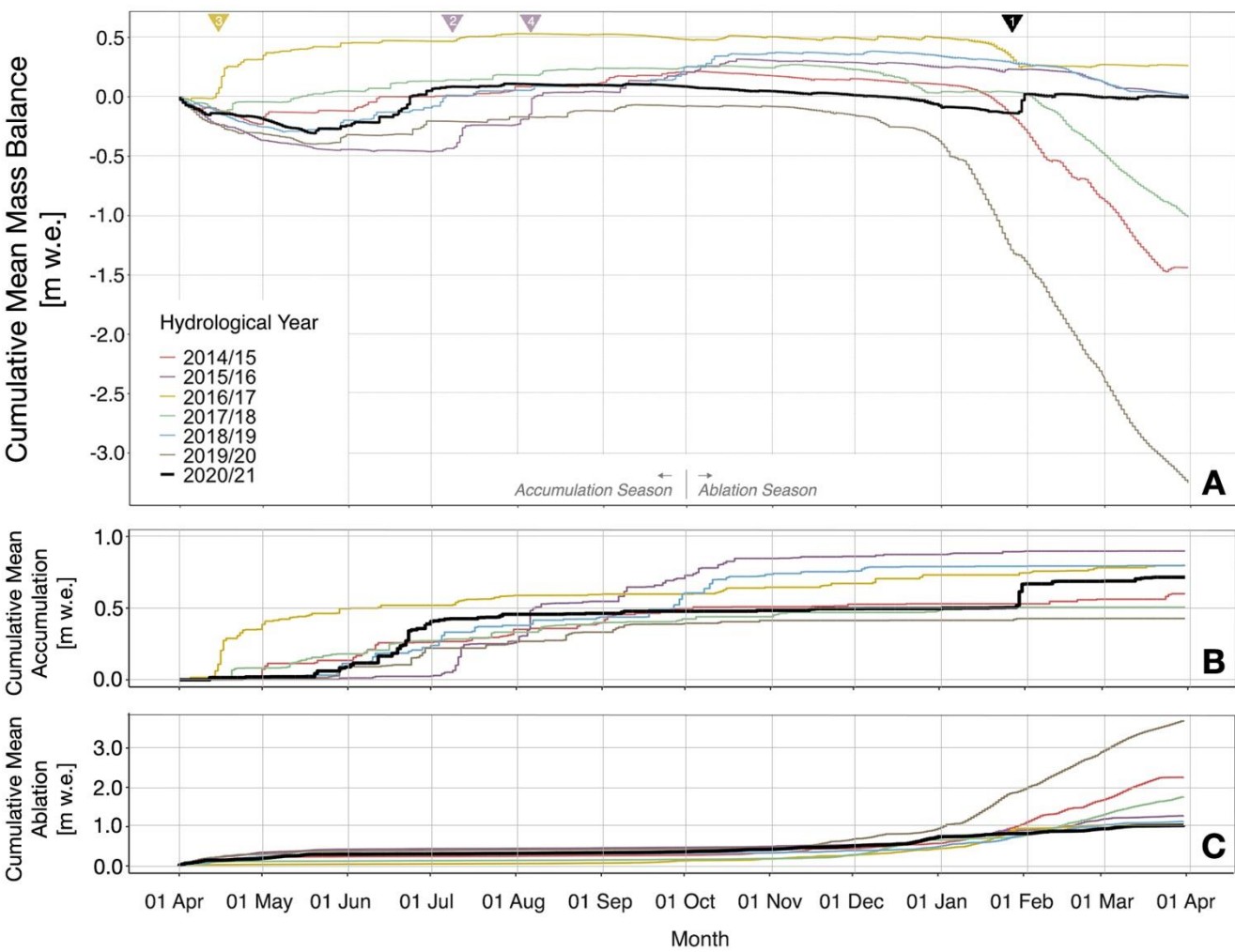

**Figure 7. a) Cumulative spatial mean mass balance, b) cumulative spatial mean accumulation, and c) cumulative spatial mean ablation of the Olivares Alfa Glacier for seven hydrological years (from 2014/15 to 2020/2021). Triangles at the top are AR events mentioned in the text with their respective categories. Note that vertical axis scales are different**


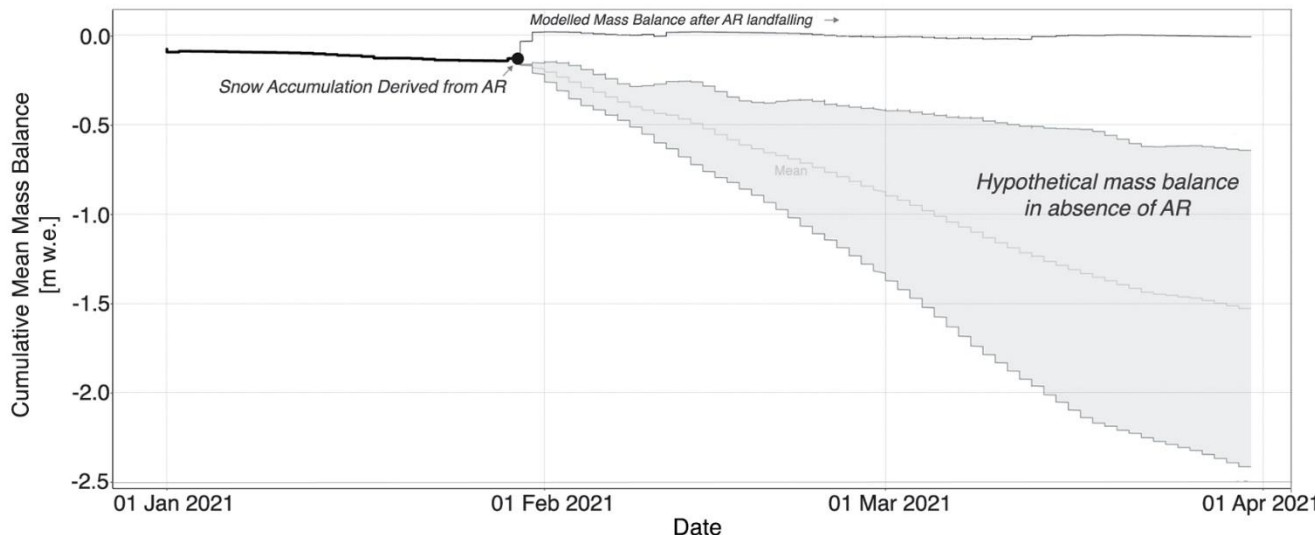

**Figure 8.** Hypothetical glacier mass balance scenarios for the 2020/21 hydrological year without occurrence of the January 2021 AR event. The graph shows a zoom for the period January to March 2021. The uncertainty range corresponds to 1σ.

## 5 Discussion

### 5.1 Modelling approach: Uncertainties and comparison with previous studies

The modelling approach incorporates several parameterizations and assumptions which may introduce uncertainty to the outputs of the model. For instance, using a constant lapse rate for air temperature is not realistic. This rate is spatially and temporally dependent on meteorological conditions (e.g., Bravo et al., 2019a). Snow albedo is highly parameterized following the snow age approach (Sauter et al., 2020; Oerlemans and Knap, 1998), while ice albedo is assumed as spatially constant when snowpack thickness is 0 m. In consequence, the ice albedo parameterization neglected ice spatial heterogeneity in a glacier where deposition of light-absorbing impurities from different sources has been detected (Barandun et al. 2022), contributing to a darkening trend of the glacier surface albedo (Shaw et al.; 2021). Moreover, small spatial heterogeneity is not captured by the model as the glacier model runs at 100 m grid size. Indeed, the presence of penitentes —spiky structures of compacted old snow and ice found on central Chilean glaciers (Lliboutry, 1954)—has been noted in the Olivares River sub-basin glaciers. A pre-event picture of Paloma Norte Glacier (Fig. S4, see Fig. 1 for location) shows the existence of penitentes. This glacier feature significantly affects energy balance fluxes by altering net shortwave and longwave radiation compared to a flat surface (Corripio and Purves, 2005). Further, despite the context of mega-drought in Chile (Garreaud et al., 2019) with almost all the years showing below-average precipitation (Fig. S5 for Lagunitas), the assumption of no snow at the start of each hydrological year could lead to an underestimation of the mass balance, since snow could persist from one year to the next.

Despite these complexities, the model's primary focus is to evaluate the relative impact of AR events on the mass balance, not to precisely replicate the exact conditions of the Olivares Alfa Glacier. Thus, the model maintains consistent parameters across each hydrological year to evaluate the general surface energy fluxes and mass balance of the glacier. Although no direct measurements of energy balance fluxes for Olivares Alfa Glacier exist, similar modelling and observational analyses have been conducted on nearby glaciers such as the Juncal Norte in the Aconcagua River Basin, as well as on the Bello and the San Francisco in the Maipo River basin (Ayala et al., 2017; Schaefer et al., 2020), all of them focusing on summer months. When comparing summertime mean values for different glaciers (Table 2), a consensus exists that net shortwave radiation and sensible heat fluxes function as sources of energy, whereas net longwave radiation and latent heat fluxes act as energy sinks.

**Table 2. Comparison of energy balance fluxes estimated in summertime by this work (Olivares Alfa Glacier) and by previous studies in nearby glaciers of central Chile.**

| Glacier | Olivares Alfa | Juncal Norte | Bello | San Francisco | Olivares Alfa (AR event 29-31 Jan. 2021) |
|---|---|---|---|---|---|
| Net shortwave radiation (W m$^{-2}$) | 120 | 70 to 285 | 223/220/208 | 137/135/149 | 22 |
| Net longwave radiation (W m$^{-2}$) | -53 | -75 to -45 | -69/-48/-69 | -42/-19/-43 | -17 |
| Sensible heat flux (W m$^{-2}$) | 18 | 0 to 65 | 25(32)/6/32 | 11(41)/6/13 | 8 |
| Latent heat flux (W m$^{-2}$) | -39 | -70 to -10 | -22(-29)/-33/-22 | -2(-9)/-5/-1 | -14 |
| Energy balance modelling | Distributed | Distributed | Point-Scale | Point-Scale | Distributed |
| Elevation (m a.s.l.) | 4284-4988 | 2904-5896 | 4134 | 3466 | 4284-4988 |
| Aspect | NE | N | SE | SE | NE |
| Period | Summer (DJF, 2014-2021) | Dec. 2008 to Feb. 2009 | Jan.-Mar. 2015 | Mar. 2016 | 28-31 Jan. 2021 |
| Source | This work | Ayala et al. (2017) | Schaefer et al. (2020)* | Schaefer et al. (2020)* | This work |

*Schaefer et al.(2020) estimated the energy fluxes using three different approaches.

Radiative and turbulent fluxes in Olivares Alfa align with those from previous studies (Table 2), despite the inherent differences due to distributed versus point-scale modelling, such as varying periods, elevation, and aspect. Juncal Norte and Olivares Alfa

glaciers share a similar northward aspect, contrasting with the predominantly southeast-facing San Francisco and Bello glaciers. Characteristics of valley topography and shadow cast effect also contribute to flux spatial variations.

Regarding mass balance, geodetic estimations for Olivares Alfa Glacier over the last twenty years are negative, ranging from -0.5 to -1.2 m w.e., with variations depending on the period (Farias-Barahona et al., 2020; Hugonnet et al. 2021). Estimation by Hugonnet et al. (2021) shows that the mean mass loss in Olivares Alfa reached -1.2 m w.e. for the period 2015-2019,

slightly higher than our estimates of -0.9 m w.e. for the same period. Our seven-year model average was -0.8 m w.e., comparable to the -0.9 m w.e. (2000-2013) and -0.8 m w.e. (2000-2019) estimates by Farias-Barahona et al. (2020) and Hugonnet et al. (2021), respectively.

The nearby glacier Echaurren Norte (0.2 km$^2$), monitored annually with the glaciological method by the Chilean Water Directorate (DGA) and reported to the World Glacier Monitoring Service (WGMS, 2023), showed a negative mass balance

with a mean of -1.9 m w.e. during the same period of this study. This difference is likely due to its elevation range (3650-3880 m a.s.l.). Its interannual variability magnitude is 1.1, versus 1.3 m w.e. for Olivares Alfa. Despite slight discrepancies of 0.1 to 0.3 m w.e. compared to geodetic balances, our modelling result is deemed effective in capturing the atmospheric condition responses and interannual variability for Olivares Alfa Glacier.

**5.2 A glacier accumulation event rather than a glacier melting event**

Despite the AR event in January 2021 being a warm midlatitude frontal storm (Valenzuela et al. 2022) and some documentation of ARs with glacier melt events in mid-latitude regions (e.g. Little et al., 2019; Kropač et al. 2021, Box et al., 2022), the January 2021 AR event in the central Andes of Chile predominantly caused snow accumulation on the glaciers in the Maipo River basin glaciers.

Several factors explain this impact. At synoptic scale, although humidity by itself is not a factor to determine if an event is

accumulation or melt, the significant moisture transport determines the magnitude of, in this case, the accumulation on the glacier surface and its imprint in the mass balance 2020/21 (Fig. 7). The maximum IVT value during this event (425 kg m$^{-1}$ s$^{-1}$) was the highest for January-February events compared to previous years (Fig. 2a, and Table S1 for the period 2014-2021), where typically IVT values during AR events oscillate between 295 and 363 kg m$^{-1}$ s$^{-1}$. Additionally, the long duration of the event contributed significantly to the remarkable larger accumulation over the glaciers in summer compared to previous years.

By comparison, AR events in winter can exhibit much higher IVT values, reaching up to 1056 kg m$^{-1}$ s$^{-1}$, like the AR category 4 event in early August 2015 (Table S1), which is also reflected in the accumulated mass balance (Fig. 7).

A primary factor to determine snow accumulation is the freezing level. At the Maipo River basin, post-event snowline altitude was below the elevation of the frontal zones of the lower glaciers which according to Ayala et al. (2020) are around 2600-2700 m a.s.l. However, although snow accumulation was predominant, we cannot rule out that melt and rain occurred in some

glaciers during the event, especially on the lower glaciers in the Maipo River basin.

Glaciers in the Olivares River sub-basin are located above 3500-3600 m a.s.l. (Fig. 1), and the freezing level during the precipitation event descended to 3200 m a.s.l. Precipitation at the beginning of the event occurred with the 0°C isotherm at

around 3600-3800 m a.s.l., suggesting positive temperatures at lower glacier sections and probably sleet-type precipitation. However, as a result, the post-event snowline altitude was around 2500 m a.s.l. This indicates snow accumulation began at
temperatures above freezing. Such disparity aligns with the variability of the rain-snow temperature threshold noted in other studies (e.g., Jennings et al., 2018). Indeed, differences of about 280 m between the snowline and the 0°C isotherm are observed during storms in the Andes at 30°S (Schauwecker et al., 2022), which can extend up to ~500 m during high-rate precipitation events. The high precipitation rate during the event occurred with the lower 0°C isotherm calculated as 3250 m a.s.l., while the snowline after the event was estimated at 2460 m a.s.l., indicating a 790 m difference. Furthermore, the 2°C isotherm,
commonly used to define the rain-snow partitioning (e.g., Koppes et al., 2011), reached a minimum altitude of 2800 m a.s.l. (Fig. 5), still above the detected snowline. Minder et al. (2011) presents an experiment to understand the difference between the 0°C isotherm and the snowline, discussing three physical processes driving this behaviour. An important conclusion is that this difference increases with increasing temperatures. Considering that zonally oriented ARs are relatively warmer storms, the difference found in our work could be attributed to this condition, compared to the more recurrent winter cold fronts.

Over the Olivares Alfa Glacier, summer snow accumulation is not unusual. To illustrate this, Figure 9 shows the hourly precipitation and its corresponding temperatures recorded in January and February between 2014 and 2021 by the GEONOR and the AWSOA (Fig. 1). Overall, hourly events larger than 1-2 mm h$^{-1}$ occur with negative temperatures, or up to 2°C, meaning that snow and sleet prevail. Events with temperatures greater than 2°C are less frequent and weak, usually in the 0.1 to 1 mm h$^{-1}$ range. These warm rain events may come from weak high-mountain convective cells, typical of the summertime
in the western slope of the Andes (Viale et al., 2014). Consequently, the AR impact is related to the snow accumulation as shown in Figure 9. Additionally, the impact of the AR resulted in no available melt energy during and almost three days after the event. During summertime, the availability of melt energy is typical in subtropical Andean glaciers, even at the elevation of Olivares Alfa Glacier (Table 2). Reduction of temperature and humidity gradients between the atmosphere and the glacier surface reduces the magnitude of the turbulent fluxes. Further, cloud cover decreases net shortwave radiation, while net
longwave radiation remains a net sink, only briefly becoming an energy source (Fig. 6). However, these conditions cannot be generalised for all the Maipo glaciers.

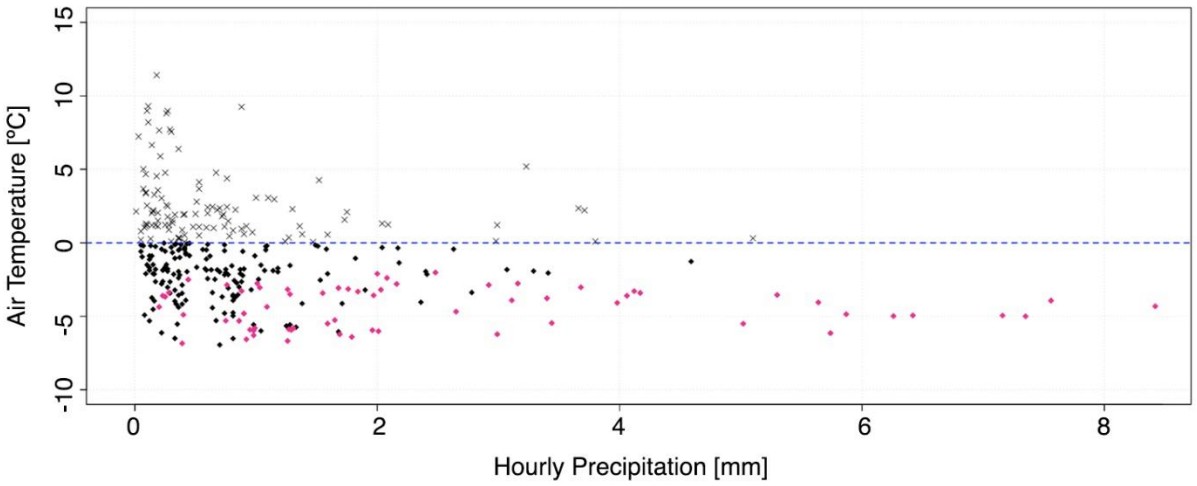

**Figure 9. Scatter-plot of the precipitation events recorded by the GEONOR sensor and air temperature recorded by AWSOA. Hourly data corresponds to January and February for the years 2014 to 2021. Crosses are precipitation events with positive air temperatures and diamonds with negative air temperatures. Pink diamonds are the precipitation rates during the event at the end of January 2021.**

Conversely, Brewster Glacier (44°S, elevation range of 1700-2400 m a.s.l.) in the Southern Alps of New Zealand experienced increased turbulent heat fluxes from the atmosphere to the glacier during a summer AR event (6 February 2011; Kropač et al. 2021), increasing the energy available for melt. The main environmental differences between glaciers in the Southern Alps and those in the subtropical Andes are related to the geographic and climatic conditions. Olivares Alfa is a high-elevation glacier with a relatively dry atmosphere, while Brewster Glacier is located at a lower elevation in a maritime environment. Both glaciers naturally experienced reduced net shortwave radiation during the summertime AR events due to cloud cover. However, while cloud cover, precipitation type and air temperature during the summer AR event contribute as energy source through the flux of incoming longwave radiation at the Brewster Glacier, they represent an energy sink at the Olivares Alfa Glacier. For the latent heat flux, the AR event's moisture input increased humidity gradient at Brewster Glacier, enhancing latent heat flux, whereas at Olivares Alfa Glacier, the atmosphere's humidity neared surface conditions, reducing the gradient. Sensible heat flux increased at Brewster Glacier due to higher atmospheric temperatures compared to the glacier surface, contrasting to the decreased flux at Olivares Alfa Glacier, where air temperature approached surface temperatures, reducing the energy available for melting. The conditions at Brewster Glacier during AR events may align closely with those observed in glaciers situated in Patagonian environments (Brown, 2020), highlighting the varied impacts of AR events based on geographic and climatic settings.

### 5.3 Seasonal conditions vs one extreme event

A regional signature of this particular AR event can be determined for all glaciers in the Maipo River basin. The mass balance record of Echaurren Norte Glacier (WGMS, 2023) shows that 2020/21 was the second least negative (-0.7 m w.e.) hydrological year since 2010, only surpassed by the 2016/17 hydrological year (-0.4 m w.e.). It is generally accepted that glacier mass balance in the subtropical Andes is influenced by the interannual variability of the snow accumulation (Masiokas et al., 2016), which is often linked to ENSO conditions, with El Niño conditions typically bringing higher amounts of snow accumulation (Cortés and Margulis, 2017). This suggests a correlation between El Niño conditions and positive glacier mass balance, as observed at Echaurren Norte Glacier (Farias-Barahona et al., 2019). Furthermore, AR frequency is anticipated to rise during the El Niño phases (Saavedra et al., 2020; Campos and Rondanelli, 2023). The 2020/21 year, however, was dominated by La Niña, notably during the summer when the AR event occurred. These conditions and our findings give insights into how an extraordinary and unseasonal snow accumulation event can significantly impact the glacier mass balance, extending beyond the usual impacts of large-scale drivers like ENSO.

As we showed, the mass balance of Olivares Alfa Glacier in 2020/21 was near equilibrium. Without the AR event, the mass balance would have been more negative, potentially rivalling or surpassing the second most negative mass balance recorded in 2014/15. This is likely according to our statistical scenarios (Fig. 8) but also supported by the higher spring snowline altitude over the past 20 years (Fig. 4) and below-average winter-spring precipitation (April to December) as recorded by the Lagunitas meteorological station (Fig. S5). However, the 2020/21 mass balance was comparable to 2015/16 and 2018/19, the two years with the highest winter accumulation during the seven years analysed in the Olivares River sub-basin. In particular, the 2015/16 year experienced two major winter AR events under El Niño conditions, one in mid-July and another between August 5 and 9 (Category 4).

The 2016/17 hydrological year had the most positive mass balance, characterised by an important accumulation event in April 2016, which indeed was an AR category 3 (Fig. 7, Table S1). After several accumulation events of lower magnitude in winter and spring 2016, the mass balance stabilised at about 0.5 m w.e. until ablation started in January 2017. Early in the year, El Niño conditions favoured AR events (Saavedra et al., 2020; Campos and Rondanelli, 2023), but the latter part shifted to neutral and La Niña conditions. Contrary to typical expectations, 2016/17 positive mass balance was during predominant La Niña conditions, significantly impacted by a single autumn AR event, illustrating how ENSO and AR events jointly influenced the 2016/17 annual mass balance.

It is important to remark that the impact is not solely from the event itself. Two small accumulation events in February and March (Fig. 7b), combined with relatively low air temperature during these months (Fig. S6) reduced the ablation rates towards the end of the hydrological year (Fig 7c).

# 6 Conclusions

In this study, we investigated the impact of an unseasonal atmospheric river (AR) and its associated unusual summertime orographic precipitation on the mass balance of the Olivares Alfa Glacier in the Maipo River basin. This summer AR transported a large amount of water vapour from lower latitudes, impacting central Chile at the end of January 2021 and producing an extraordinary and unseasonal orographic precipitation event in the subtropical Andes (Valenzuela et al 2022). By utilizing remote sensing, meteorological in-situ observations, and glacier modelling, we thoroughly assessed the impacts of this AR-driven precipitation event. A lower-than-normal 0°C isotherm and a snowline below the glacier's elevation range were detected during and after the event, indicating predominant snow accumulation across the glaciers of the Olivares River sub-basin, and potentially the entire Maipo River basin's glaciers.

Modelling of the Olivares Alfa Glacier shows that the high unseasonal snow accumulation had a significant role in altering the annual mass balance. This glacier mass accumulation event abruptly interrupted the ongoing summertime melting trend for the rest of the ablation season. Surface energy balance analysis shows that melting energy was absent during the event, mainly due to the reduction of the typical source of energy in summer, i.e., net shortwave radiation and sensible heat flux (Table 2). Although energy sinks, such as net longwave radiation and latent heat flux, saw a reduction in their magnitude, they did not offset the decrease in energy sources.

As a consequence, the 2020/21 annual balance was near equilibrium ($\sim$0 m w.e.), despite the high spring snowline and below-normal winter precipitation typical of La Niña conditions. Further, a decrease in the ablation rate and the occurrence of small snow accumulation events in the period between after the event and until the end of the hydrological year contribute to the near equilibrium. In the context of seven simulated hydrological years (from 2014/15 to 2020/21), the annual mass balance of the 2020/21 year resembled those seen in last El Niño years, underscoring how an extraordinary accumulation event can offset expected responses to large-scale climatic drivers like ENSO.

At a global scale ARs are projected to become more frequent in a warming climate (Nellikkattil et al., 2023) and intense in terms of their precipitation rate (Wang et al., 2023). Further, Li and Ding (2024) found a poleward shift in the occurrence of ARs during the boreal winter between 1979 and 2022, increasing the occurrence south of 50°S and decreasing the occurrence north of 30°S. Given the impact of unseasonal snow accumulation tied to ARs, such as the January 2021 event, on glacier mass balance, we anticipate a growing influence of such events on Andean glaciers south of 30°S, particularly when they occur out of season. Moreover, in addition to causing unseasonal snow accumulations, ARs have the potential to trigger significant melting events. The occurrence of unseasonal extreme precipitation events, whether they lead to accumulation or melting, will introduce new challenges in our way of analysing glacier mass balance data, necessitating enhanced in-situ campaigns, more comprehensive mass balance observations, and expanded modelling efforts to better understand the spatiotemporal variability and impacts of such extreme events on glacier dynamics.

**Author contributions**

CB, SC and MV designed the outline of this study, PP prepared the weather station data and calculated 0°C isotherm, NGL prepared the ERA5 data, SC run the model and calculated snowline, MV catalogued the historic atmospheric river events. CB, SC, MV, PP and NGL contributed with the figures. CB and SC analysed the data and write the manuscript. DB and MV provided guidance on results interpretation. MV, NGL and DB reviewed and edited the paper. All authors contributed to the final paper.

**Competing interests**

The authors declare that they have no conflict of interest.

**Acknowledgments**

We acknowledge the providers of the data used in this work: Copernicus Climate Change Service (C3S) Climate Data Store for the ERA5 reanalysis data, Integrated Global Radiosonde Archive (IGRA) Version 2 for radiosonde data and National Snow and Ice Data Center for the MODIS daily snow cover product. We thank the team at Centro de Estudios Científicos (CECs) who installed, downloaded the data and carried out the AWS and GEONOR maintenance between 2013 and 2021. We are grateful to the editor, Emily Collier, and Alvaro Ayala and one anonymous reviewer for their detailed revision, constructive comments and suggestions.

**Funding**

This research has been supported by the Agencia Nacional de Investigación y Desarrollo (ANID) through the program FONDECYT Iniciación 11240379 and by CECs. MV is supported by FONCYT 2020-1722. DB acknowledges support from ANID-FONDAP-1523A0002 and COPAS COASTAL ANID FB210021.

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
