# Peer review of "Unseasonal atmospheric river drives anomalous summer snow accumulation on glaciers of the subtropical Andes"

_EGUsphere, 2024_

## Author Comment (AC1)

**Author Response:**

**We appreciate the constructive review and comments by Anonymous Review 1. We agree with the major comments and we will introduce changes in the manuscript to address the reviewer's concerns. Also, we will clarify and correct the manuscript considering the specific comments. We think that these changes will improve the manuscript. Here, we provide a brief point-by-point response to the general and specific comments and concerns by the Anonymous Reviewer 1 (in bold):**

Summary

In this paper, Bravo and coauthors study an unusual July 2021 atmospheric river (AR) that contributed positively to the mass balance of glaciers in the subtropical Andes during what is normally the ablation season. They use a combination of station observations, atmospheric reanalysis, remote sensing data, and a glacier mass balance model to show that this AR halted the seasonal progression of Olivares Alfa Glacier mass loss, resulting in near-equilibrium mass balance for the year. They conclude that a single major AR event can exert a dominant influence on annual glacier mass balance, even when the large-scale climate conditions would normally be expected to favor mass loss.

The paper presents a compelling scientific story and is a novel contribution to the literature, as AR impacts on the cryosphere have not been studied in detail in this region of the subtropical Andes. The paper is generally well-written with sound scientific methods, and the references are comprehensive and appropriate. The main aspect the paper lacks is a more thorough exploration of how this AR event compares to the long-term climate context of this study region, as described in my major comments. I also have a large number of minor comments and technical corrections that do not represent fundamental flaws, but should be addressed before the paper can be of a publishable standard.

Major comments

- Since the ERA5 data are available for a longer time period than 2014–2021, it would enhance this study to see how the January 2021 compares to all summer ARs during the multi-decadal ERA5 record. I understand that running the COSIPY model for this long time period is likely beyond the scope of this analysis, but I don't expect it would be too difficult to extend the record of AR events and their categories to the full time period of the ERA5 data. This would provide some valuable long-term background to determine how unusual the January 2021 AR was. Some of this information may be provided at the regional scale by previous studies (e.g. Valenzuela et al., 2022), but the long-term context that is directly relevant to the glacier mass balance of this study area should be provided and interpreted for the reader in this paper. See also my minor comment on L356–358.

**We thank the reviewer for this suggestion. We will add an analysis to emphasise the extraordinary occurrence of this event. For one side, we will use ERA5 to catalogue the Atmospheric Rivers in summer to demonstrate the historic low occurrence of this synoptic feature. Effectively, as the Reviewer mentioned, a longer mass balance modelling using for instance ERA5 data, is beyond the scope of this work as we focus on feeding the model using the available meteorological observations.**

- How does the total accumulated precipitation compare to past AR events, both during

summer and during all seasons? Is there precedent for this type of summer accumulation event if you look at a longer time period than 9 years? Is there any way for the authors to quantify this with the available data? I expect that the record-high IVT values relative to the Jan/Feb 2013–2021 climatology (L354–361) would translate to precipitation accumulation at the high end of the climatology, but this isn't guaranteed to be the case.

**In the same line as the previous answer, we will add an analysis to show how the rate of precipitation was also extraordinary for summer. In this case, we will use available data observed at Lagunitas weather station. This station, although located at a lower elevation (2765 m a.s.l.), shows one of the longest precipitation records, so we can put this event in the context of an observed precipitation climatology. We will replace Figure 3 with a Figure to remark this. Regarding a comparison, we mention that the IVT is not the highest in the period, as winter ARs events showed the highest IVT. The January 2021 event was the one with the highest IVT in summer since 2014. We expected from the analysis of the extended AR catalogue (previous answer) to determine if this event was the one with the highest IVT on summer in a longer period.**

- Figure 3: It's not clear why this figure is included in the paper. The only references to this figure in the text are to mention that the station data exist (L175, 177) and the figure is not used to support any of the paper's main findings. It should either be removed from the paper, or some text should be added to the paper describing how the figure contributes to the study's results. See also my minor comment on L424–425.

**We agree with this comment. However, the Figure will be moved to Supplementary Material as we still want to show and remark on the importance of this high-elevation meteorological observation that has been recorded data since 2013. Also, we estimate that this figure still is important to demonstrate the validation of the ERA5 incoming longwave radiation used to feed COSIPY as, unfortunately, not all the period has available data of this variable.**

Minor comments

- L32–45: This is a long paragraph. I suggest starting a new paragraph at L38 with the sentence starting with "In the subtropical Andes..."

**We agree. We will start a new paragraph at L38.**

- L38: It would be helpful to give some more detail about what region the term "subtropical Andes" refers to. What countries / areas of countries are considered the subtropical Andes? L42 implies that this mainly refers to Chile and Argentina, but it would be useful to define this region in more detail at its first mention in L38.

**Generally, subtropical Andes encompassed the area between south of the tropic until around the 40°S, however, in this paragraph, we refer to the area between 32° and 36°S. We will add a sentence to clarify this.**

- L88 (last paragraph of introduction): This paragraph jumps abruptly to discussing the January 2021 AR event without any transition from previous paragraphs. It would be helpful to at least briefly discuss the seasonal climatology of precipitation and ARs in this region as context for the January 2021 AR studied in this paper. This type of information is

provided to some extent later in the paper (e.g. L128–129), but it would be helpful for developing the paper's story for it to be included in the introduction.

**We will move a couple of sentences from L128-129 and add some climatology context using the work of Viale et al. (2018). The paragraph will be:**

*"According to Viale et al. (2018), in the subtropics, ARs are much more frequent in winter. Further, precipitation and ARs are almost absent in summertime (dry season) over the western slopes of the subtropical Andes and the central Chilean lowlands (Viale and Garreaud, 2014; Viale eat al., 2018). Despite these overall characteristics, intense precipitation occurs in summer (Poveda et al., 2020). In this work, our objective is to evaluate…"*

- Figure 1: A large-scale map, showing the study region's location within the broader context of southern South America, would be helpful for readers unfamiliar with the region's geography.

- Figure 1: Do the blue areas on the large map show the outlines of glaciers? Please clarify.

**Thank you for these observations, we will add a broader regional context and also the location of Lagunitas weather station (following a specific comment below). Blue areas do represent the glacier areas on the basin, we will detail this information in the figure caption.**

- L110–141: I suggest reorganizing this section to cover only the study area. I think the paper will flow better if the description of the January 2021 AR event is refactored into the Introduction and Results sections. Any background information on the January 2021 AR that is based on previous studies (e.g. Valenzuela et al., 2022) should be moved to the Introduction, and any analysis of this event that is a new result of this study should be moved to the Results.

**Given the major comments, the new analysis aimed to remark on the extraordinary of this event will be presented in the Results section. The background information in this section will be moved to the Introduction.**

- L122: Is there any way to label the two accumulation zones of the Olivares Alfa Glacier on the large map in Figure 1?

**We will add this detail to the map in Figure 1.**

- L131–132: The climatology of AR category 1 events during summer in this region is helpful context for the reader. Is summer defined as December-January-February? Please clarify.

**Yes, it is DJF, we will add this information. This section was now moved to the Introduction.**

- Figure 2: It would be helpful to mark the location of the Maipo and/or Olivares River basin on either panel A or C.

- Figure 2: It doesn't make sense to me to have a combined y-axis with IVT magnitude on the left axis (units of kg m^-1 s^-1) and IVT direction (angular units) on the right axis. I suggest splitting this into two separate panels with different y axes.

**We will make the corrections to this Figure as the Reviewer suggested.**

- L152–169: These methods for determining the snowline elevation and freezing level are a nice, creative blending of remote sensing, radiosonde, and station data.

**Thank you for your comment.**

- L153–154: Is this product, which I presume is based on visible imagery, affected by cloud cover? Are there places / times where the snowline elevation can't be determined due to clouds?

**The method employed relies on the MODIS snow product (MOD10A1), which is known to be susceptible to cloud cover, limiting its accuracy in areas with frequent cloud coverage. However, the approach proposed by Krajčí et al. (2014) overcomes this limitation by offering improved tolerance to clouds. By allowing a higher cloud cover threshold —90% in our case— this method can still provide reliable snow cover information, even in cloud-prone regions. This enhanced cloud tolerance results in better accuracy compared to other snow detection methods, such as those by Parajka et al. (2010) and Da Ronco and De Michele (2014), which perform less effectively under heavy cloud cover.**

- L179–180: From what source(s) are the medium and high resolution imagery?

**We will add a table in the supplementary material showing the satellite images used to outline the glacier. Overall these are Landsat, Spot6, Kompsat3 and Pleiades images.**

- L183–184: Do the authors anticipate that initializing the model with a no-snow starting condition for each year will have an influence on the results? It would be helpful to include at least a brief discussion of the implications of this decision.

**During the study period, the subtropical Andes of Chile has been affected by an extensive mega-drought. Most of the years has shown a deficit of precipitation. On the other side, MODIS derived snowline shows higher elevation at the end of the hydrological years, even over the maximum elevation of the Olivares Alfa Glacier. Moreover, experienced in fieldworks in the area since 2013, shows that during summer most of the snow deposited on the glacier was melted leaving the ice exposed at the end of each hydrological year. With this in mind, we decided to initialize the model with no snow to maintain consistent parameters across the study period (L341-342), however, we are aware that some snow could exist at the start of the hydrological year in the highest elevations of the glacier, especially the hydrological year following years with highest rate of precipitation (see Figure S3).**

**Considering this, it is probably that the mass balance of these years will be a bit higher. We will add this in the Discussion section 5.1. as a source of uncertainties.**

- L189–195 and L296–299: I like the idea of simulating a hypothetical scenario for seasonal mass balance evolution without the AR's influence, but I'm not sure I completely follow the method and the conclusions that can be drawn from it. How was the detrending of the mass balance time series post-event performed? Am I interpreting L193–195 correctly to mean that this method isn't capable of assessing the influence of the albedo increase during the AR event?

**Thank you for your comment. We will clarify this statistical method. We decided on this approach because we discarded the influence of the feedback related to the albedo increase post-event as well as the feedback related to other variables during the event such as incoming shortwave and longwave radiations, wind speed, relative humidity, atmospheric pressure and air temperature. Effectively it is possible to run COSIPY assuming no precipitation during the days of the events, but the event itself also forces other meteorological variables impacting mass balance. An option would be to create an artificial time series for each variable, but we decided on a statistical approach so we could obtain a range of hypothetical values of mass balance.**

- L208–221: Is this analysis of ARs during 2014–2021 for all seasons? Or summer only? Please clarify.

**We will clarify this. It is for all seasons.**

- L231–232: I'm not sure I agree with the statement that the snowline elevation did not return to its pre-event elevation by the end of the hydrological year. If I am interpreting Figure 4 correctly, it looks like the snowline returned to its pre-event elevation by early March, then another snow event in mid-March decreased the snowline elevation once again.

**Actually, the day previous to the event the snowline was 4693 m a.s.l. and the mean of the 10 days previous to the event was 4654 m a.s.l. After the event and until the end of the hydrological years these values were not reached again. The maximum value was on 25 march (4493 m a.s.l.) and the mean was lower.**

- L249–251: Do the authors have any hypotheses for why there was a greater discrepancy in the 0 degree isotherm from radiosondes vs in-situ temperature sensors during the post-event days? Is there a physical reason for why this might be the case, or is it just a random occurrence?

**Checking the HYSPLIT model (attached Figure), the 29 and 30 of January show that the radiosounding launched from Santo Domingo was in the direction to the Andes, while the 2 of February shows a trajectory to the north-west, to the Pacific. We hypothesise that this difference in trajectories determines the discrepancy. Anyway, the good match between radiosonde and observations probably responds to the**

**dominant synoptic conditions of the days of the event, despite the direction of the radiosonde is not exactly over the Olivares basin.**

[Figure]

- Figure 4: This is a nice plot that does a good job of illustrating the radiosonde and station comparisons. However, I have a couple of comments on this plot:

- Similar to my comment on Figure 2, I suggest splitting the plot into 2 separate panels with separate y-axes, rather than having two different scales on the same y-axis

- The two rightmost x-axis tick marks are incorrectly labeled as January. These dates are in February.

**Thank you for your comments and our apologies for the wrong labelled marks, we will change this.**

- L258–274: Be clear that the energy fluxes reported in this section are based on the COSIPY model simulation rather than observations. This is discussed in Section 5.1 but this point should also be made clear here.

**We agree and we will add a clarification.**

- L282: I don't see the support for the statement that "Typically, ablation in April dominates the mass balance". If I am interpreting Fig. 7 correctly, it looks like the largest mass loss months in the 2014–2021 COSIPY simulations were February and March.

**We agree with the reviewer that the statement is not clear. We refer that overall, along the study period, April of the different years shows a predominance in the ablation processes over accumulation, not that the rate of ablation is larger than in the other summer months. To clarify, we will start this sentence: "*Typically, ablation in April is larger than accumulation…*"**

- L288–289: This sentence states that "As expected, the ablation season started in September 2020", but the x-axis label in Fig. 7 labels Oct 1 as the start of the ablation season.

**We agree with the reviewer's comment. Actually, the ablation starts earlier than expected. To avoid confusion, we will edit this sentence, adding "*early than expected…*"**

- L337: Nice job compiling the estimates of glacier mass balance from previously published sources and comparing them with the COSIPY simulations. This lends credibility to the study results.

**Thank you for your comment.**

- L356–358: Be clear that the record of historical January-February events, to which the IVT value is being compared, covers only the period from April 2013 to March 2021 (according to Table S1).

**We will add the requested information. Is compared with January and February between 2014 to 2021.**

- L366–376: This is an interesting discussion of the discrepancy between the observed snowline and the height of the 0 degree isotherm. Do the authors have any hypothesis for why the snowline was anomalously low relative to the 0 degree isotherm during this event?

**We don´t have a hypothesis regarding this discrepancy observed here and also in the Andes at 30°S (Schauwecker et al., 2022), however Minder et al. (2011) present an experiment to understand the difference between 0°C isotherm and the snowline. From this experiment, three physical processes are discussed as responsible for this behaviour. An important conclusion is that the difference increases with increasing temperatures. Considering that ARs are relatively warmer storms, the difference found in our work could be explained by this condition in comparison to the more recurrent cold fronts. We will reference Minder et al. (2011) in this section.**

- L410–429: This is a nice discussion of how one extreme event can counteract the evolution glacier mass balance expected from the large-scale climate state. This is a good story for the reader to take away from the paper.

**Thank you for your comment.**

- L424–425: This appears to be an erroneous reference to Fig. 3. Fig. 3 does not say anything about how the snowline elevation has changed over the past 20 years.

**We apologise for this mistake.**

- L424–426: Where is the Lagunitas meteorological station located? This station should be shown on a map in one of the figures, and also described in Section 3 (rather than introducing this dataset for the first time near the end of the paper).

**We will add the location to the map of Figure 1. Also, we will introduce it early, considering that we will use this data to remark on the extraordinary rate of precipitation of this even in summer (mayor comment).**

- L463–464: The two papers referenced in this sentence describe projected future changes

in global AR conditions. Are there any references that provide projections that are more directly relevant to the study region? Or do the two referenced papers include results that can be used to describe projections more specifically for this study region?

**Both studies remark as main conclusion the increase of AR and precipitation associate to AR at global scale. Specifically, the study area seems to be in the limit where increase of AR and extreme precipitation associated to is projected. This agree with the results of a recently published work that shows an increase in AR south of 50°S and a decrease north to 30°S (Li and Ding, 2024). However, this analysis it is just for the boreal winter**

Technical corrections

**We are very grateful for the technical corrections by the reviewer. We will introduce all the changes suggested by the Reviewer and rephrase some of the sentences. We apologize for the typos and erroneous grammar.**

- L20: "over" --> "in"

- L37: "are" --> "is"

- L39: insert "the" before El Niño

- L50: "role of glaciers" --> "influence on glacier"

- L53: "its" --> "their"

- L60: "mid-latitudes" --> "mid-latitude"

- L64: "ARs" --> "AR"

- L65: "its" --> "their"

- L73: "over" --> "above"

- L90: "and characterized" --> "and was characterized"

- L90: "mountain" --> "mountains"

- L96: "the well-known large-scale glacier mass balance forcing as ENSO" - this does not make grammatical sense, please rephrase

- L136: "fuel" --> "fueled"

- L145: IVT stands for "integrated vapor transport" or "integrated water vapor transport", not "integrated vertical transport"

- L147: "Ralph's scale" - please rephrase

- L147: Remove the word "current"

- L150: "scales" --> "scale"

- L159: "was" --> "were"

- L212: "into" --> "over"

- L226: "glaciers" --> "glacier"

- L227: "at this summertime" - this phrase does not make grammatical sense, please rephrase

- L244: "increases" --> "increase"

- L246: "similar values of" --> "similar values to"

- L257: "Surface fluxes energy" --> "Surface energy fluxes"

- L262: "nigh" --> "night"

- L280: "are" --> "is"

- L285: It is not clear what "particular" means here. Please choose a different word.

- L354: "At synoptic-scale, significant moisture transport." - This sentence is a fragment, please revise.

- L367: Remove the word "up"

- L367: "occurs" --> "occurred"

- L394–395: "heat turbulent" --> "turbulent heat"

- L396: "Glaciers" --> "glaciers"

- L401: "influx longwave radiation" - This phrase does not make grammatical sense, please rephrase.

- L453: "mass glacier" --> "glacier mass"

- L461: "202/21" --> "2020/21"

**New references**

Minder, J. R., Durran, D. R.  and Roe, G. H.: Mesoscale controls on the mountainside snow line, J. Atmos. Sci., 68, 2107–2127, https://doi.org/10.1175/JAS-D-10-05006.1, 2011.

Li, Z., and Ding, Q.,A global poleward shift of atmospheric rivers. Sci. Adv.10, eadq0604(2024). https://doi.org/10.1126/sciadv.adq0604, 2024.

---

## Author Comment (AC2)

**Author Response:**

**We appreciate the constructive review and comments by Dr. Alvaro Ayala. We agree with the major comments and we will introduce changes in the manuscript to address the reviewer's concerns. Also, we will clarify and correct the manuscript considering most of the specific comments and suggested Figure editions. We think that these changes will improve the manuscript. Here, we provide a brief point-by-point response (in bold) to the general and specific comments and concerns by Dr. Ayala:**

PAPER SUMMARY AND RECOMMENDATION

Bravo et al. analyse the impact of an unseasonal atmospheric river (AR) on the annual mass balance of Olivares Alfa Glacier, subtropical Andes of Chile. The AR occurred at the end of January 2021and resulted in a strong precipitation event over central Chile, which is very rare to occur during the austral summer. The authors conducted their analyses using remote sensing products, meteorological observations, and energy and mass balance models. They found that the event produced an accumulation of 164 mm w.e. (measured near the glacier tongue) and lowered the 0°C isotherm from typical summer elevations of 4000-4500 m a.s.l. to 3000-3500 m a.s.l., as well as lowering the snowline elevation to about 2500 m a.s.l. Glacier mass and energy balance modelling shows that the annual mass balance of Olivares Alfa Glacier was close to neutral as a consequence of the AR. Synthetic simulations indicate that without the event the annual balance of Olivares Alfa would have been very negative (between -0.5 and -2.5 m w.e., approximately).

The topic of the article is novel and appropriate for The Cryosphere. The analyses seem adequate, and the main message is interesting and useful for future studies. I suggest that the authors add a few more analyses and clarifications to make the article ready for publication.

MAJOR COMMENTS

1. How rare was this event on glaciers?

I agree with the main comment of reviewer 1. I understand that Valuenzuela et al. (2022) showed a detailed analysis on a regional scale, but it would be useful to know how often such an accumulation event occurs on glaciers in the study area. Can you add some more analysis in this direction? Calculate a return period from Lagunitas data? Or maybe add data from ERA5 and El Yeso meteorological station?

**We thank the reviewer for this suggestion in line with a Major comment of Reviewer 1. We will add an analysis to emphasise the extraordinary occurrence of this event and its accumulation. For one side, we will use ERA5 to extend the catalogue of the Atmospheric Rivers in summer to demonstrate the historic low occurrence of this synoptic feature. Also, we will add an analysis to show how the precipitation rate was also extraordinary for summer. In this case, we will use available data observed at Lagunitas weather station. This station, although located at a lower elevation (2765 m a.s.l.), shows one of the longest precipitation records so that we can put this event in the context of an observed precipitation climatology. We previously performed this analysis (see Figure attached, in Spanish) but we didn't include it in the manuscript. We will do it in the potential new draft. Events with almost 100 mm in Lagunitas occur**

**between 35 to 65 years. In the figure, the red asterisk is the January 2021 event and the analysed period is 1960-2024.**

[Figure]

2. Mechanisms that explain the mass balance change

The authors state that "… the impact is not solely from the event itself. Feedback mechanisms related to snow accumulation also impact the mass balance. After the event, ablation diminished due to reduced surface temperatures and increased albedo, which lowered net shortwave radiation, which is the main source of energy for melting during summer" (lines 438-440). So, which was more important? It would be good to answer this question very clearly in the abstract and conclusions. I see that the total snow accumulation at the location of AWS was 164 mm w.e. (Fig. 5) and that the expected ablation without the event ranges between -500 to -2500 m w.e. (Fig. 8). Can you conclude that the main effect of the event was to change the energy balance rather than the mass gain during the event? If this is the case, I think it could be stated more clearly.

I have other suggestions along these lines that could help to understand the effect of the storm on the glacier energy and mass balance.

- Figure 6: Can you add two more panels showing i) albedo and ii) surface temperature? It would be interesting to know for how long the albedo remained high.
- Satellite images: Can you show satellite images to better understand how the AR affected the glacier surface during the rest of the summer? For example, I can see from a Sentinel image of 09.03.2021 that the glacier was already quite dark on that date, but a few days later, a small snowfall brought the albedo back to high values again. So, maybe there were other snowfalls that contributed to keeping the mass balance neutral by increasing the albedo?

**Checking the albedo outputs of COSIPY, we agree that the post-event feedback is not directly related to the event. As is parametrized, the albedo reduces quickly,**

which seems to agree with Reviewer's comments on the satellite images. The low ablation rate in the last two months of the hydrological year seems to be related with two smaller events of snow accumulation in February and March and also to relatively lower air temperature on these months (Fig. S4). The magnitude of these events is not unusual in summer (see Figure below), but it forced a similar impact on the albedo. With this in mind, we will change our statement, and concentrate on the importance of the 164 mm w.e. is quite extraordinary for the date if compared with previous years (e.g. Lagunitas precipitation data) and this was due to the AR. Also, we mentioned that the albedo parametrization is a source of uncertainty (L312-317), to be consistent we will not discuss in detail the post-event albedo. However, we will add in the discussion section that the two accumulation events impacted, reducing the ablation. Just briefly, both, albedo and surface temperature during February are lower than the previous years for the same month, but we recognize this is not solely by the AR event but also for other events. An analysis using satellite images is beyond the scope of this work.

[Figure]

3. Hypothetical scenario ("no event")

This is a very interesting and useful exercise, but the description provided by the authors is very brief. What were the main assumptions made? What were the time series of precipitation, temperature and the other variables that you used? The same as those recorded, except for precipitation? Is surface albedo calculated by the model? How low would have been the glacier albedo without the event?

Figure 8: Can you add a panel showing the albedo in the actual and hypothetical scenario? What was the effect of the small events after the AR on surface albedo?

**The approach here was statistical, using the time series of the mass balance of similar behaviour years (L190-195). Therefore, no time series of air temperature and other meteorological variables. We decided on this approach because we discarded the influence of the rest of the variables during the event such as incoming shortwave and longwave radiations, wind speed, relative humidity, atmospheric**

pressure and air temperature. Following also comment of Reviewer 1 we will add more details to clarify this approach. We will provide more details about this procedure as follows: "The mass balance time series from previous years were decomposed to extract the trend for each year (Box et al., 2015). Then, the 2020-2021 mass balance series was detrended, and the average, maximum, and minimum trends derived from previous years, in terms of the final mass balance result, were applied to the analysed hydrological year."

MINOR COMMENTS

Title: I think that the title is not fully accurate. "Glacier accumulation" is not the most common term. Maybe change to "snow accumulation", "glacier mass accumulation", "glacier snow accumulation" or "glacier mass gain"? E.g. "Unseasonal atmospheric river drives anomalous summer snow accumulation on glaciers of the subtropical Andes".

**Thank you for your suggestion. We will modify the title.**

Data availability: Are the meteorological data going to be available?

**In the short-term, by request.**

21: "… led to substantial snow accumulation on the Maipo River glaciers, confirmed by the post-event snowline …" I don't think that the low snowline confirms a substantial snow accumulation, because a cold event with low precipitation can also produce a low snowline.

**We agree, we will change to: "… led to substantial snow accumulation on the Maipo River glaciers and post-event snowline observed at …"**

58-62: Can you briefly explain how an AR could produce more melt? Is it because it rains on the glaciers? I thought that an AR was always associated with a precipitation event.

**AR originate in the intertropical zone. Therefore, both, water vapour and high temperature are transported poleward by the AR, are transferring to the glacier as energy available for melt. According to Kropač et al. (2021), this is through longwave radiation and strong turbulent heat fluxes. Rain heat flux also plays a role in fuelling melt.**

112: The 70% number is originally from a DGA report, maybe check if there is a more recent number? Maybe Álvarez-Garretón or CR2 have calculated a more updated number in recent years.

**We will update this. Although is not the same, we agree that is more relevant to mention that 60% of the water of the basin is used in the agricultural sector and 35% is for drinking water and sanitation.**

122: I'm not sure if "two accumulation zones" is technically correct. Maybe say that the accumulation zone is divided in two valleys or cirques.

**We agree.**

159: Can you show the sensors along the Olivares Basin on a map?

**We will add the location of the air temperature sensor in the new map of Figure 1.**

181: Can you briefly explain how the model distributes meteorological variables? Precipitation, winds? How is snow and ice albedo calculated by the model? What value did you use for ice albedo? From observations or the literature?

**These steps are explained in the paper by Sauter et al. (2020). Briefly, the model used lapse rates for air temperature, and relative humidity. The barometric formula for atmospheric pressure and modelling approaches for shortwave and longwave radiation. Wind speed is constant. For albedo, the Oerlemans and Knap (1998) approach is used, assuming theoretical values of 0.3 for ice and 0.85 for fresh snow.**

191: "detrending the mass balance time series post-event" This is not clear, how was this procedure? Can you provide more details about this experiment? Did you remove all the summer precipitation? Is albedo adjusted by the model? See my major comment 3.

192: "The behavior from previous similar years … was derived and applied to the detrended 2020/2021 accumulated mass balance time series" I don't follow the procedure. I thought that the experiment consisted only of running the model without the AR event, but did you use information from other years? See my major comment 3.

**191-192: We answered this in the major comment. We hope this clarifies the method. We added a reference.**

260: The negative latent heat flux means in this context sublimation, not melt. What happened to the snow deposited by the event? Was it sublimated or melted? Can you provide both amounts? Looking at figure 6, I would say that sublimation dominated over melt after the event.

**We will correct this statement; it is not clear. After the event, the snow melted and a small fraction sublimated. The figure below shows that there is sublimation over the hydrological year but the rate is lower than the melt. During the event and in the other summer events, no melt is registered but sublimation continues.**

[Figure]

**FIGURES**

Figure 1: A, please delete the rest of the political boundaries, or explain what they are. The text refers only to the Maipo River Basin.

**We will edit this Figure.**

Figure 3: Please change the red colour of the ERA5 longwave radiation. It is difficult to distinguish from the black lines. Maybe change this plot from hourly to daily time steps? As it is, the hourly data have a lot of noise.

**Following the comment of Reviewer 1, we will move this Figure to the Supplementary section. We will keep the hourly time step because is the time step that we used for feeding the model.**

Figure 4: Can you indicate the event period here?

**We will add a marker to indicate the period of the event.**

Figure 5: -> "Time series of the 0°C isotherm around the event"

**We will correct this.**

Figure 5: What is AWS DGA? So, you didn't use the Ta sensors along the valley to calculate the isotherm?

**We will change this. We used several air temperature sensors as is described in the manuscript.**

Figure 5: The number 164.6 mm w.e. is only given here, and it is quite important. Please mention it also in the text.

**Ok, thank for noting this omission. We will introduce this information in the text.**

Figure 6: Please see my main comment 2.

Figure 8: Can you add another panel showing accumulation and ablation separately? I think that would be very useful to understand whether the cause of the neutral mass balance was the snow accumulation during the event or its effect on surface albedo.

**We will add what the reviewer suggested. Similar to the Figure below.**

[Figure]

Table 2: Can you add a new column with the average fluxes in the days or weeks after the event? This would make it easier to understand the changes caused by the AR (instead of looking at Figure 6).

**We will add a column with the mean values during the event.**

SUGGESTED TECHNICAL CORRECTIONS

**We are very grateful for the technical corrections by the reviewer. We will introduce all the changes suggested by the Reviewer.**

18: add "austral" to "summer"

20: -> "the effects of the AR on the…"

21: Replace "significant" by another term, maybe "massive" or "large".

25: Introduce the current mega-drought before or maybe just say "a severe drought". As it is, the sentence assumes that all readers know about the prevailing mega-drought conditions.

35: Delete "during specific periods, such as the hydrological year"

36: -> "there is a typically large interannual variability"

138: -> "strong even for winter events"

150: This sentence is quite orphan. Remove or move to the introduction. Or provide here some more general details.

209: "Pacific coastal grid points" Refer to Figure 2c.

211: Please move "Category 1 being the lowest and …" to line 131 when the categories are first mentioned.

213: -> "by the amount of time" or maybe "duration"

231: "an elevation like January 2021", which one?

239: "Diurnal cycle" is more precise

250: precise here if the direction of the discrepancy, what is higher and what is lower?

383-385: But this is logical, no? It is the ablation season.

415: "Cortés and Margulis"

425: I think it should be Fig. 4, not 3.

**New References**

Alvarez-Garreton, C., Boisier, J. P., Garreaud, R., González, J., Rondanelli, R., Gayó, E., and Zambrano-Bigiarini, M.: HESS Opinions: The unsustainable use of groundwater conceals a "Day Zero", Hydrol. Earth Syst. Sci., 28, 1605–1616, https://doi.org/10.5194/hess-28-1605-2024, 2024.

Box, G. E. P., Jenkins, G. M., Reinsel, G. C., and Ljung. G. M.: Time Series Analysis: Forecasting and Control (5th ed.), Wiley, United States, 720 pp., ISBN 978-1-118-67502-1, 2015

---

## Referee Report (RR1)

Review for egusphere

**Title: Unseasonal atmospheric river drives anomalous summer snow accumulation on glaciers of the subtropical Andes**

Authors: Bravo, Cisternas, Viale, Paredes, Bozkurt, García-Lee.

**PAPER SUMMARY AND RECOMMENDATION**

Bravo et al. analyse the impact of an unseasonal atmospheric river (AR) on the annual mass balance of Olivares Alfa Glacier, subtropical Andes of Chile. The AR occurred at the end of January 2021and resulted in a strong precipitation event over central Chile, which is very rare to occur during the austral summer. The authors conducted their analyses using remote sensing products, meteorological observations, and energy and mass balance models. They found that the event produced an accumulation of 164 mm w.e. (measured near the glacier tongue) and lowered the altitude of the 0°C isotherm from typical summer values of 4000-4500 m a.s.l. to 3000-3500 m a.s.l., as well as lowering the snowline to elevations of about 2500 m a.s.l. The authors propose that the annual mass balance of Olivares Alfa Glacier was close to neutral because of the AR and other factors. A statistical analysis based on results for previous seasons indicate that without the event the annual balance of Olivares Alfa would have been negative (between -0.5 and -2.5 m w.e., approximately).

This is the second version of an article that I already revised with positive comments. I had three major comments. I am satisfied with the responses provided by the authors except for a few details that need more clarification in the article. I also have a second round of suggested text changes to improve clarity and flow.

**UPDATE ON MY MAJOR COMMENTS**

    1. How rare was this event on glaciers?

Thanks for the new analyses. The exceptional occurrence of the event is now much more evident.

    2. Mechanisms that explain the mass balance change

In the revised version, the authors state that (496-498) "It is important to remark that the impact is not solely from the event itself. Two small accumulation events in February and March (Fig. 7b), combined with relatively low air temperature during these months (Fig. S5) reduced the ablation rates towards the end of the hydrological year (Fig 7c)".

I appreciate that the authors have extended the possible causes that explain the near-neutral mass balance to causes other than the accumulation from the AR. Please add a similar (maybe shorter) statement in the Conclusions and Abstract. I don't want to reduce the impacts of your findings, but I find these considerations very important.

Moreover, I think it must be clear in the article that, although the event was remarkable and with large impacts across Central Chile, the snow accumulation during the event (0.16 m w.e.) was relatively small in comparison to the typical annual ablation (between 1 and 4 m, Figure 7c). This is why I think that apart from the accumulation (which I agree is remarkable for summer), the increase in surface albedo due to the fresh snow, plus the small events that came later, plus the low temperatures are all responsible for the neutral mass balance.

    3. Hypothetical scenario ("no event")

Thanks for explaining this exercise more in detail. I see now more clearly what the analysis was, but I think that the methods require a few more clarifications:

212-216: "To further evaluate the impact of this unseasonal precipitation event, we **also simulated** the 2020/21 hydrological year's mass balance under a hypothetical scenario without the AR's influence, removing the accumulation attributed to the summer-2021 AR. To complete the last two months of the 2020/21 hydrological year, the mass balance time series from previous similar years (i.e., **those years with negative mass balance until the end of January**) were decomposed to extract the trend for each year (Box et al., 2015). Then, the 2020-2021 mass balance series was detrended, and **the average, maximum, and minimum trends** derived from previous years, were applied to the analysed hydrological year to hypothesize a scenario range without the occurrence of the AR."

- "we also simulated": To be fair, you didn't perform new simulations, you use the results from previous years. Please rephrase as it now suggests that you used COSIPY to perform new simulations.
- "those years with negative mass balance until the end of January": How many years are those, only three? Please mention them explicitly.
- "the average, maximum, and minimum trends": Figure 8 shows values of "sigma", is that a standard deviation calculated from only three curves? Did you use standard deviation or a minimum and maximum?

**MINOR COMMENTS**

- Use of the word "rate"

The article uses the word "rate" many times, but I think that it is sometimes superfluous and in other times misleading. Please check the use of this word throughout the article. It seems that many times the authors refer to precipitation totals and not really "a rate", in the sense that can be understood as an hourly or daily rate. I would say that the important feature of the article was the amount of precipitation and not really a rate, or do the authors refer to a particular hourly or daily rate?

- Figure 6

Do these fluxes correspond to the location of the AWS or are averaged over the entire glacier? My guess is that they correspond to the location of the AWS because the snow seems to disappear very quickly. If that is the case, then please reword the caption to: "Glacier energy balance fluxes estimated by COSIPY at the location of the AWS."

46: citations?

46-48: This knowledge gap contradicts what you asserted in the previous sentence, maybe reword?

48: significance -> magnitude and frequency

144: Historical satellite images -> from 1955? Those would be aerial and not satellite.

248: Citations for this sentence?

252: Is the fact that the AR in January 2021 is a Category 1 somehow unexpected? In the Introduction you mention that Category 1 are mainly beneficial.

446: Why the conditions can't be generalised?

481-482: I don't agree with this sentence. I think that the cold conditions that prevailed during the rest of the summer would have prevented the mass balance to be so negative as in the other years.

**FIGURES AND TABLES**

Figure 4: Can you indicate the event period here?

Figure S1: Please change the red colour of the ERA5 longwave radiation. It is difficult to distinguish from the black lines.

Table 2: Define the event period in the caption

**SUGGESTED TEXT CHANGES**

22: glaciers -> basin?

33: Glaciers are highly sensitive to climatic variations and stand as…

34: Glacier mass balance is a metric that is central to their behaviour and is a critical …

36: Long-term observations, over what period?

37: Delete "However"

37: Embedded in this overall negative mass balance trend there is a large interannual variability in atmospheric conditions that is transferred to the annual mass balance of glaciers.

40: I suggest to reword starting with "The atmospheric interannual variability in the subtropical Andes of Chile and Argentina (between 32S and 36S)…"

42: the Southern Annular Mode….

43: delete "of the subtropical Andes"

44: sometimes -> can

45: modulating annual mass balance -> driving general trends in glacier mass balance

46: modulate -> impact

56: Delete "Notably"

58: Delete "semi-arid"? it is a bit confusing jumping from semi-arid to sub-tropical

59: "about 50%"?

61: "on mid-latitude glaciers" and delete "mid-latitude glaciers" at the end of the sentence.

63: dynamics -> mass balance

65: Suggest rearranging: "…showed that the days with the most intense ablation and largest accumulation rates at Brewster Glacier…."

71: Suggest rearranging to eliminate the commas, maybe: "Conversely, ARs can trigger substantial snow accumulation events on glaciers located above the 0C isotherm."?

73: "mid-latitudes, but can also exert a …"

80: notably contributing to glacier albedo changes -> largely increasing the albedo

82: in -> on

83 semi-arid -> sub-tropical? Or define both regions? I would try to stick to one only.

85: dynamics -> mass balance

90: enhances -> can help to enhance

90: Central Andes? This could be confusing too.

91-93: Please check the wording an rearrange for a better readability. Viale et al (2018) is cited at the beginning and again at the end.

94: Delete "For example", this is the main event of the article

96: During the event, precipitation in central-southern Chile exceeded…

100: Join both paragraphs

104: marking the first study analysing the impact of an AR on the mass balance of glaciers in the sub-tropical Andes.

106: contextualise the extraordinariness -> describe the exceptional characteristics

106: unusually large precipitation

133: sanitization -> sanitation?

133: The basin contains…

134-135: Our analyses focused on the the Olivares Ricver sub-basin (coordinates here), which hosts…

136: These glaciers have experienced the largest ice loss… ("most significant" is not technically correct)

141: they have shown -> their surfaces have shown a trend towards darkening since …

142: We simulate the energy and mass balance of the Olivares….

145: continuous -> larger

146: delete all after "2013", or why does the loss of area directly indicate fragmentation?

151: why "extend"? maybe "build a catalogue"? This is the first time that a catalogue is mentioned in the article.

151: "a catalogue of the summertime AR in the Andes"

152: rate or total? Or did the event have a high daily or hourly precipitation rate?

153: "we identify AR conditions on the grid points of the ERA5 reanalysis data representing the coast of Central Chile in the 1941-2023 period".

160: "of at least Category 1 conditions that occurred"?

163: Please check the verb tenses of all the sections. It seems that this section is in present tense, but the next section is written in Past tense.

163: of AR -> the AR

164: delete "on"

165: the available 83-year period

173: Firstly, we estimate if glaciers in the Maipo River Basin were …"

174: "For this, we utilised…"

180: Delete "Furthermore"

183: "1a) to estimate the freezing level."

183-184: "The freezing level was estimated using a linear regression…"

185: "accuracy and reliability" of what?

193: Please provide the simulation period at some point at the beginning of the paragraph.

195: not only turbulent fluxes, I guess, but the full energy and mass balance

196: "To cover the complete simulation period, we derived data before 2016 from the ERA5 reanalysis, after going…"

198: "incoming shortwave and longwave radiation"

202: delimitations -> outlines?

203: "The model distributes…" This fits better somewhere in the previous paragraph.

206: "annual approach" is not a very clear term, maybe is better just explain?

207: no-snow starting -> snow-free initial

208: "The analysed model outputs"

208: total -> glacier total

210: was made with that of previous…

212: Please start here a new paragraph

232: Please check the use of the word "rate"

233: "costal grid points of Central Chile"

234: rarity -> low frequency?

236: the total number of AR events during all seasons reached 687

239: delete "nearby"

241: Move "Valenzuela et al. (2022)…" sentence to the end of the paragraph.

250: at Lagunitas station

265: "ERA5 reanalysis"

267: 19 AR summertime (DJF) events

275: usually being

277: it -> this value

279: January 2021 -> Maybe more like December 2020?

286: the 0C isotherm altitude

287: isotherm altitude estimates

289: minimum altitude of the 0C

290: "Around this time, … were accumulating snow."

293: at the glacier lowest elevations

294: up -> down?

294: "decreased to altitudes similar to those observed in the previous day"

296: "On the days after", How many?

298: radiosonde-derived

302: Time series of the 0C isotherm (left axis) and hourly precipitation (right axis) during the event

304: Bars indicate

304: correspond

307: indicate that the

309: Latent heat fluxes are predominantly negative (), indicating that…

312: daily mean value?

315: in the magnitude of both the net shortwave and longwave

319: decrease of the surface temperature

321: why does the persistent cloud cover reduce the temperature gradient?

326: remove "according to…"?

328: remove "and the accumulation and ablation"

329: May-March?

330: wit the mass balance of 2019/20

330: The mass balance in 2018/19 was

330: while the mass balance in 2016/17

331: with a positive value

331: maybe add "cumulative" somewhere, to make sure that you don't refer only to April values. Also, it would be better to say that is "the trend" that is interrupted, I think.

336: The mass balance in 2020/21 shows

339: earlier than

345: please reword, the glacier experienced the mass loss, not the years.

347: a sensitivity -> a hypothetical

348: delete "negatively"

352: Cumulative glacier mass balance, or why "mean"? is that a spatial average?

352: delete all "mean"

353: are -> indicate, represent

357: Hypothetical glacier mass balance

362: using a constant lapse rate for air temperature is not realistic.

364: The explanation of the snow albedo parameterization should on "Methods"

364: ice albedo is assumed as spatially uniform and spatially constant.

365: the ice albedo parameterization

367: of the glacier surface

368: … penitentes – spiky…- has been noted on the Olivares River sub-basin glaciers.

370: "Indeed", repetition

372: context of a severe drought

375: the relative impact

377-378: What do you that no direct measurements of energy balance fluxes are available? The other studies use probably the same type of data (AWSs)

387: those from previous studies

388: such as varying periods, elevation,

391: Geodetic mass balance of Olivares Alfa Glacier for the last 20 years has been negative.

393: Can you add the uncertainty of the Hugonnet estimates? I guess it is quite a lot for such a short period

397: Mention somewhere that the monitoring of Echaurren is done in-situ using the glaciological method

397: Not only is reported to WGMS, but is also the reference glacier for the Southern Andes

406: snow accumulation on

408-410: Can you reword using less commas? It would help the flow of the reading

410: The maximum IVT value? Or the mean?

412: the long duration

413: rate of snow accumulation -> snow accumulation values compared to previous summers

416: determine the occurrence of snow accumulation on glaciers

435: summer snow accumulation is not unusual. To illustrate this, …

441: "after the event", for how long?

42: , even at the elevation of …

443: the glacier surface

445: However, these conditions…

47: GEONOR sensor

451-453: Please check the sentence structure

459: of incoming longwave radiation

459: they represent an energy sink

461-464: Please reword. It is not very clear.

475-475: Same here. Please consider splitting the sentence.

480: The mass balance would have been

485: Maybe delete "during 1991-2021", it confuses a bit the sentence

485: Delete "rate"

501: orographic summertime precipitation

503: delete "in the subtropical Andes"

508: has -> had

511-512: The reduction in the magnitude of energy sinks did not compensate for the decrease in energy inputs.

519-521: Please reword, not very clear with so many commas.

521: accumulations -> accumulation

527-530: Check the verb tenses (all in present)

---

## Author Response (AR2)

**We appreciate the constructive review and comments by Anonymous Reviewer 1 and Dr Alvaro Ayala. We introduce most of the change and suggestion by both reviewers and tried to address the reviewer's concerns. We kept some sentences despite the suggestions as we consider are related with different styles in redaction. Also, we corrected the manuscript considering the specific comments. We think that these changes improved the manuscript. Here, we provide responses to the comments and technical issues detected by the reviewers (in bold):**

**Anonymous referee #1**

Summary

Thanks to the authors for their thorough responses to the reviewer comments. The analyses of ERA5 ARs and Lagunitas precipitation added in Section 4.1 now provide important long-term context for the January 2021 case study event. Figure 1 is improved from the previous version and is now more useful for understanding the study region.

I do have a number of remaining minor comments and technical corrections, as detailed below. Several of the minor comments are requests for the authors to incorporate helpful information from the reviewer response into the manuscript itself. Provided these comments are addressed, in my assessment this paper is on the right track toward publication.

Follow-ups on previous comments

Note: Unless otherwise noted, all line numbers in these follow-up comments refer to the previous version of the manuscript, in order to preserve the linkage with previous comments and the reviewer response.

- Major comment 2: I was hoping that the analysis of the Lagunitas precipitation data would also give an estimate of the return period of the Jan 2021 event compared to the all-season climatology, in addition to the summer climatology. However, I don't think this is a major issue, and will leave it to the authors' discretion whether they wish to include this analysis in the next version.

**The comparison of the dry-season AR2021 PRecip event with all-seasons precip events is not relevant. For all-seasons climatology, the precipitation accumulated in the summer AR2021 PP event is not rare. But it occurs in the dry season, which is extremely dry in the subtropical Andes (only 10% of the annual total see Viale and Garreaud 2014) and glacier normally do not accumulate mass.**

- Figure 1: Please clarify in the caption that the blue areas are glaciers, as promised in the reviewer response. These could be mistaken for lakes.

**Added**

- Figure 3 (previously Figure 2): Please mark the location of the study area on panel a and/or c (I suggest both), and add text to the caption noting this.

**Added.**

- L153–154: The response to this comment in the reviewer response is helpful for understanding the MODIS snowline data, and would be helpful for readers in addition to reviewers. Please incorporate this text and references into the manuscript itself.

**We added part of the answer to the manuscript.**

- L179–180: I do not see where any information about the satellite image sources has been added to the supplementary material.

**We added the Table in the Supplementary Material. Sorry for the omission.**

- L231–232: These specific values of snowline elevation before and after the event are helpful for making the authors' case about the impact of the AR on mass balance. Please incorporate this text from the reviewer response into the manuscript itself.

**We added part of the answer to the manuscript.**

- L249–251: This is an interesting hypothesis about the different directions the radiosondes likely traveled during versus after the event. I suggest adding the HYSPLIT figure from the reviewer response as a supplementary figure, and adding a sentence to the end of Section 4.3 that incorporates some of this explanation from the reviewer response.

**We added the Figure as supplementary material and part of the answer.**

- Figure 5 (previously Figure 4): I still think this would be better as a 2-panel figure, with the panels vertically aligned so that the temporal correspondence can still be seen between the high snow accumulation rates and the lower of the 0C isotherm. But I do not feel strongly enough to press the issue further and will leave this to the discretion of the authors.

**We understand the observation of the Referee but we prefer to keep it as is, as we look to show the direct relation between precipitation and 0°C isotherm. We think is not so much information for one panel.**

- L463–464: Please incorporate some of the text from the reviewer response - about AR projections specific to the study region - into the manuscript itself.

**We added part of the answer to the manuscript and the new reference.**

Other comments

- Upon re-reading the paper, I feel it is still slightly lacking in insight about the interaction between the zonally-oriented AR and the regional topography that produced the orographic precipitation. Looking at Fig. 1 and at Google Maps for wider context, it appears that the Lagunitas weather station and the Olivares River sub-basin are separated by a ridgeline to the northwest of the Olivares River sub-basin, and there are is also a north-south oriented mountain crest to the west of the Olivares River sub-basin that may block moisture transport from the west. I can also see from Google Maps that the main topographic divide separating Chile and Argentina lies to the east of the sub-basin. I understand that Lagunitas is the nearest station with a long-term record, but I think a better picture of the regional topography would be helpful in interpreting the observations

from this station in relation to conditions on the Olivares Alfa Glacier. I think what would probably be helpful would be to add DEM shading, and shade the ocean in blue, in the bottom inset panel of Fig. 1a to help the reader get their bearings and visualize the interaction of westerly flow with the topography. I apologize for the vague and rambling nature of this comment, I just feel that something is a bit missing.

**Thank you for the comment. We kept the Figures as is, as the contour level gives the information about the topography.**

- This is a very minor comment, but in the Fig. 1 caption, it would be helpful to point out that the Lagunitas weather station is located near the top left of the large map in Fig. 1a. I had trouble finding the location of the station at first glance, and confused it with the magenta "T5" AWS location marker.

**Added.**

- L306–324: I am a bit confused by these statements about the melt energy. It looks like the melt energy (black dashed line in Fig. 6) was mostly flat with the exception of a few days (25 Jan, 03 Feb, 04 Feb). Does "typically reaches a mean value of 54 W m-2" refer to the long-term average of melt energy for this time of year? And does the maximum around 150 W m-2 just before the AR event refer to the brief spike in melt energy on 25 Jan?

**Thank you for your comment. We clarify in the manuscript that 54 W m-2 refers to the long-term mean and that the maximum of 150 wm-2 observed before the events is a typical peak at this time of the year, so it works as an example.**

- L451: I suggest changing "Southern Alps" to "Southern Alps of New Zealand" here. I know the location of Brewster Glacier is mentioned in the introduction, but since this section abruptly switches the geographical context from South America to New Zealand, it would be helpful to remind readers where Brewster Glacier is located.

**Added.**

- L494: Does "this year" refer to 2016/17 or to 2020/2021?

**We reworded this sentence and delete confuse sentence.**

Technical corrections

**All the technical correction suggested by the Reviewer were include in the new version.**

- L23: "typically" --> "typical"

- L40: "between the" --> "between"

- L59–60: Does this sentence mean that AR-related snowfall events are 2.5 times more intense than non-AR snowfall events? Please clarify.

**Yes, we clarified this**

- L103–104, 116–117: Both these paragraphs have the redundant statement that this is the first AR-glacier impact study in this region. I suggest removing this statement from the sentence in L103–104.

**Changed as suggested.**

- L107: "extraordinariness" --> "extraordinary nature"

- L141: "darkening their surfaces" --> "surface darkening"

- L143: "divide" --> "divided"

- L151: "historically" --> "climatological"

- L157, 158: "AR" --> "ARs"

- L160: "events at least" --> "events if at least"

- L163: "measure of" --> "measure"

- L164: "evaluating on" --> "evaluating"

- L167: "remark" --> "remark on"

- L170: "in the highlands of this Andes" --> "in this region of the Andes" (?)

- L239: "series" --> "time series"

- L254: The precipitation map is shown in Fig. 3c, not Fig. 3b

- L265: "Reanalysis ERA5" --> "ERA5 reanalysis"

- L265, L489, and elsewhere: After the abbreviation "AR" has been defined in the main text, be consistent with using the abbreviation rather than the full term "atmospheric river" thereafter.

- L273: "75th percentile" --> "the 75th percentile"

- L275: "at" --> "during"

- L277: "Immediately" --> "Immediate"

- L285–299 and check elsewhere: Be consistent with writing "0°" vs "0°C"

- L298: "radiosonde derived" --> "the radiosonde-derived"

- L286: "of January" --> "January"

- L286: "0°C" --> "the 0°C"

- L304: "corresponding" --> "correspond"

- L305: "2°C" --> "the 2°C"

- L306: "fluxes" --> "flux"

- L311: "their" --> "the"

- L315: "the 29" --> "29"

- L322: "the 30" --> "30"

- L322: "fluxes" --> "flux"

- L330: "the 2016/17" --> "2016/17"

- L331: "7-years" --> "7-year"

- L339: "early" --> "earlier"

- L345: "at" --> "during"

- L345: "their" --> "the"

- L408–410: This sentence is confusingly worded. Please rephrase.

**We delete a part to avoid confusion.**

- L418: "discard" --> "rule out"

- L430: The 2°C isotherm is shown in Fig. 5, not Fig. 4

- L472: "amount" --> "amounts"

- L480: "the equilibrium" --> "equilibrium"

- L484 and elsewhere: Be consistent with how hydrological years are expressed (e.g. 2020/21 vs 2020/2021)

- L505: glacier's --> the glaciers'

- L508: "has" --> "had"

**Referee #2 Dr. Alvaro Ayala**

Review for egusphere

Title: Unseasonal atmospheric river drives anomalous summer snow accumulation on glaciers of the subtropical Andes

Authors: Bravo, Cisternas, Viale, Paredes, Bozkurt, García-Lee.

PAPER SUMMARY AND RECOMMENDATION

Bravo et al. analyse the impact of an unseasonal atmospheric river (AR) on the annual mass balance of Olivares Alfa Glacier, subtropical Andes of Chile. The AR occurred at the end of January 2021and resulted in a strong precipitation event over central Chile, which is very rare to occur during the austral summer. The authors conducted their analyses using remote sensing products, meteorological observations, and energy and mass balance models. They found that the event produced an accumulation of 164 mm w.e. (measured near the glacier tongue) and lowered the altitude of the 0°C isotherm from typical summer values of 4000-4500 m a.s.l. to 3000-3500 m a.s.l., as well as lowering the snowline to elevations of about 2500 m a.s.l. The authors propose that the annual mass balance of Olivares Alfa Glacier was close to neutral because of the AR and other factors. A statistical analysis based on results for previous seasons indicate that without the event the annual balance of Olivares Alfa would have been negative (between -0.5 and -2.5 m w.e., approximately).

This is the second version of an article that I already revised with positive comments. I had three major comments. I am satisfied with the responses provided by the authors except for a few details that need more clarification in the article. I also have a second round of suggested text changes to improve clarity and flow.

UPDATE ON MY MAJOR COMMENTS

1.

How rare was this event on glaciers?

Thanks for the new analyses. The exceptional occurrence of the event is now much more evident.

2.

Mechanisms that explain the mass balance change

In the revised version, the authors state that (496-498) "It is important to remark that the impact is not solely from the event itself. Two small accumulation events in February and March (Fig. 7b), combined with relatively low air temperature during these months (Fig. S5) reduced the ablation rates towards the end of the hydrological year (Fig 7c)".

I appreciate that the authors have extended the possible causes that explain the near-neutral mass balance to causes other than the accumulation from the AR. Please add a similar (maybe shorter) statement in the Conclusions and Abstract. I don't want to reduce the impacts of your findings, but I find these considerations very important.

Moreover, I think it must be clear in the article that, although the event was remarkable and with large impacts across Central Chile, the snow accumulation during the event (0.16 m w.e.) was relatively small in comparison to the typical annual ablation (between 1 and 4 m, Figure 7c). This is why I think that apart from the accumulation (which I agree is remarkable for summer), the increase in surface albedo due to the fresh snow, plus the small events that came later, plus the low temperatures are all responsible for the neutral mass balance.

**Thank you for your comments. We add this information in the new version of the manuscript and now also remark this fact in the Conclusion.**

3.

Hypothetical scenario ("no event")

Thanks for explaining this exercise more in detail. I see now more clearly what the analysis was, but I think that the methods require a few more clarifications:

212-216: "To further evaluate the impact of this unseasonal precipitation event, we also simulated the 2020/21 hydrological year's mass balance under a hypothetical scenario without the AR's influence, removing the accumulation attributed to the summer-2021 AR. To complete the last two months of the 2020/21 hydrological year, the mass balance time series from previous similar years (i.e., those years with negative mass balance until the end of January) were decomposed to extract the trend for each year (Box et al., 2015). Then, the 2020-2021 mass balance series was detrended, and the average, maximum, and minimum trends derived from previous years, were applied to the analysed hydrological year to hypothesize a scenario range without the occurrence of the AR."

-

"we also simulated": To be fair, you didn't perform new simulations, you use the results from previous years. Please rephrase as it now suggests that you used COSIPY to perform new simulations.

**Our approach here was statistical, that´s why we used the word simulation. However, following Reviewer observation in terms that it seems that we perform new COSIPY runs (which is not) we changed this phrase.**

"those years with negative mass balance until the end of January": How many years are those, only three? Please mention them explicitly.

**Added**

"the average, maximum, and minimum trends": Figure 8 shows values of "sigma", is that a standard deviation calculated from only three curves? Did you use standard deviation or a minimum and maximum?

**Minimum and maximum. We deleted the sigma in the Figure to avoid confusion.**

MINOR COMMENTS

-

Use of the word "rate"

The article uses the word "rate" many times, but I think that it is sometimes superfluous and in other times misleading. Please check the use of this word throughout the article. It seems that many times the authors refer to precipitation totals and not really "a rate", in the sense that can be understood as an hourly or daily rate. I would say that the important feature of the article was the amount of precipitation and not really a rate, or do the authors refer to a particular hourly or daily rate?

**We check this and change when necessary following Reviewer suggestion.**

Figure 6

Do these fluxes correspond to the location of the AWS or are averaged over the entire glacier? My guess is that they correspond to the location of the AWS because the snow seems to disappear very quickly. If that is the case, then please reword the caption to: "Glacier energy balance fluxes estimated by COSIPY at the location of the AWS."

**Is the spatial mean, we added this in the caption.**

46: citations?

46-48: This knowledge gap contradicts what you asserted in the previous sentence, maybe reword?

**We add citations and reword, we eliminate "individual" as the citations added mention the influence of several extreme events along the year and how this modulate the mass balance (e.g. Poveda et al., 2020, section "Southern Tropical Andes"). Then we mention that the individual impact has not be assessed.**

48: significance -> magnitude and frequency

**Changed**

144: Historical satellite images -> from 1955? Those would be aerial and not satellite.

**Changed**

248: Citations for this sentence?

**Added**

252: Is the fact that the AR in January 2021 is a Category 1 somehow unexpected? In the Introduction you mention that Category 1 are mainly beneficial.

**It is expected, as we mentioned other AR with higher category occur mainly in winter. This AR anyway is characterized because of the unseasonality but the magnitude it is lower that winter AR**

446: Why the conditions can't be generalised?

**Basically to different characteristics as surface conditions (e.g. debris) and location (e.g. shadow effects, elevation). Also, some areas are more prone to summer convective storm as Olivares, while other not.**

481-482: I don't agree with this sentence. I think that the cold conditions that prevailed during the rest of the summer would have prevented the mass balance to be so negative as in the other years.

**We deleted the word "significantly" but after the "no event scenario" we demonstrate that without the event mass balance would be negative anyway.**

FIGURES AND TABLES

Figure 4: Can you indicate the event period here?

**We added in the caption more detail about the snowline during the event.**

Figure S1: Please change the red colour of the ERA5 longwave radiation. It is difficult to distinguish from the black lines.

**Changed**

Table 2: Define the event period in the caption

**Added in the Table.**

SUGGESTED TEXT CHANGES

22: glaciers -> basin?

**We keep "glaciers", which is the focus of the manuscript.**

33: Glaciers are highly sensitive to climatic variations and stand as…

**We kept the original redaction**

34: Glacier mass balance is a metric that is central to their behaviour and is a critical …

**We kept the original redaction**

36: Long-term observations, over what period?

**We changed "long-term" for "Over the last two decades"**

37: Delete "However"

**Deleted**

37: Embedded in this overall negative mass balance trend there is a large interannual variability in atmospheric conditions that is transferred to the annual mass balance of glaciers.

**We deleted "typically" we kept the resto of the sentence.**

40: I suggest to reword starting with "The atmospheric interannual variability in the subtropical Andes of Chile and Argentina (between 32S and 36S)…"

**We kept the original redaction**

42: the Southern Annular Mode….

**Added**

43: delete "of the subtropical Andes"

**Deleted**

44: sometimes -> can

**Changed**

45: modulating annual mass balance -> driving general trends in glacier mass balance

**We kept the original redaction**

46: modulate -> impact

**We kept "modulate" as we use "impact" in the next sentence**

56: Delete "Notably"

**Deleted**

58: Delete "semi-arid"? it is a bit confusing jumping from semi-arid to sub-tropical

**Agree**

59: "about 50%"?

**Added**

61: "on mid-latitude glaciers" and delete "mid-latitude glaciers" at the end of the sentence.

**Changed**

63: dynamics -> mass balance

**Changed**

65: Suggest rearranging: "…showed that the days with the most intense ablation and largest accumulation rates at Brewster Glacier…."

**Changed**

71: Suggest rearranging to eliminate the commas, maybe: "Conversely, ARs can trigger substantial snow accumulation events on glaciers located above the 0C isotherm."?

**We think what you suggest change a bit the meaning of the sentence, so we kept it as is.**

73: "mid-latitudes, but can also exert a …"

**We kept the original redaction**

80: notably contributing to glacier albedo changes -> largely increasing the albedo

**We kept "changes" as under some conditions, a reduction in albedo could occur. For instances rain-on-snow events.**

82: in -> on

**Thank you.**

83 semi-arid -> sub-tropical? Or define both regions? I would try to stick to one only.

**Yes, sub-tropical is used.**

85: dynamics -> mass balance

**Changed**

90: enhances -> can help to enhance

**Changed**

90: Central Andes? This could be confusing too.

**Agree, we changed by "Subtropical Andes"**

91-93: Please check the wording an rearrange for a better readability. Viale et al (2018) is cited at the beginning and again at the end.

**Viale et al (2018) deleted at the beginning of the sentence.**

94: Delete "For example", this is the main event of the article

**Deleted**

96: During the event, precipitation in central-southern Chile exceeded…

**Added**

100: Join both paragraphs

**Done**

104: marking the first study analysing the impact of an AR on the mass balance of glaciers in the sub-tropical Andes.

**We reworded this sentence**

106: contextualise the extraordinariness -> describe the exceptional characteristics

**We changed "contextualize" by "describe". The rest was kept it.**

106: unusually large precipitation

**Added**

133: sanitization -> sanitation?

**Changed**

133: The basin contains…

**Changed**

134-135: Our analyses focused on the the Olivares Ricver sub-basin (coordinates here), which hosts…

**We kept the original redaction**

136: These glaciers have experienced the largest ice loss… ("most significant" is not technically correct)

**Agree, changes as suggested.**

141: they have shown -> their surfaces have shown a trend towards darkening since …

**We reworded this sentence**

142: We simulate the energy and mass balance of the Olivares….

**We kept the original redaction**

145: continuous -> larger

**We kept continous**

146: delete all after "2013", or why does the loss of area directly indicate fragmentation?

**Malmros et al. (2016) quantified a significant glacier area loss of 63% between 1955 and 2013 and indicating a gradual fragmentation of the ice mass over the years.**

151: why "extend"? maybe "build a catalogue"? This is the first time that a catalogue is mentioned in the article.

**Changed by "to catalogue"**

151: "a catalogue of the summertime AR in the Andes"

**Previous answer**

152: rate or total? Or did the event have a high daily or hourly precipitation rate?

**Changed, although rate is also defined "by unit time"**

153: "we identify AR conditions on the grid points of the ERA5 reanalysis data representing the coast of Central Chile in the 1941-2023 period".

**We kept the original sentence.**

160: "of at least Category 1 conditions that occurred"?

**We reworded this sentence adding "if"**

163: Please check the verb tenses of all the sections. It seems that this section is in present tense, but the next section is written in Past tense.

**Done**

163: of AR -> the AR

**No change here**

164: delete "on"

**No change here**

165: the available 83-year period

**Added**

173: Firstly, we estimate if glaciers in the Maipo River Basin were …"

**We kept the original sentence.**

174: "For this, we utilised…"

**Added**

180: Delete "Furthermore"

**Deleted**

183: "1a) to estimate the freezing level."

**Added**

183-184: "The freezing level was estimated using a linear regression…"

**Added**

185: "accuracy and reliability" of what?

**Of the estimation of the freezing level.**

193: Please provide the simulation period at some point at the beginning of the paragraph.

**It is indicated in the paragraph.**

195: not only turbulent fluxes, I guess, but the full energy and mass balance

**Changed as suggested**

196: "To cover the complete simulation period, we derived data before 2016 from the ERA5 reanalysis, after going…"

**We kept the original sentence.**

198: "incoming shortwave and longwave radiation"

**Added**

202: delimitations -> outlines?

**Chnaged**

203: "The model distributes…" This fits better somewhere in the previous paragraph.

**Moved**

206: "annual approach" is not a very clear term, maybe is better just explain?

**"approach" changed by "time-steps". The explanation is in the next sentence**

207: no-snow starting -> snow-free initial

**Changed "no" by "free"**

208: "The analysed model outputs"

**Changed**

208: total -> glacier total

**Changed**

210: was made with that of previous…

**Added**

212: Please start here a new paragraph

**We thing is part of the same overall idea associate to "simulation" We kept the original form.**

232: Please check the use of the word "rate"

**Changed by "Accumulated precipitation"**

233: "costal grid points of Central Chile"

**We kept Pacific.**

234: rarity -> low frequency?

**Changed**

236: the total number of AR events during all seasons reached 687

**Changed**

239: delete "nearby"

**We kept it**

241: Move "Valenzuela et al. (2022)…" sentence to the end of the paragraph.

**We kept original paragraph structure**

250: at Lagunitas station

**Added**

265: "ERA5 reanalysis"

**Changed**

267: 19 AR summertime (DJF) events

**Reworded**

275: usually being

**Added**

277: it -> this value

**Changed**

279: January 2021 -> Maybe more like December 2020?

**We reworded this section following recommendations of Referee 1.**

286: the 0C isotherm altitude

**Changed**

287: isotherm altitude estimates

**Added**

289: minimum altitude of the 0C

**Added**

290: "Around this time, … were accumulating snow."

**Changed**

293: at the glacier lowest elevations

**Changed**

**294: up -> down?**

**Deleted**

294: "decreased to altitudes similar to those observed in the previous day"

**Reworded**

296: "On the days after", How many?

**Not changed**

298: radiosonde-derived

**Changed**

302: Time series of the 0C isotherm (left axis) and hourly precipitation (right axis) during the event

**Reworded**

304: Bars indicate

**Added**

304: correspond

**Changed**

307: indicate that the

**Added**

309: Latent heat fluxes are predominantly negative (), indicating that…

**Changed**

312: daily mean value?

**Reworded**

315: in the magnitude of both the net shortwave and longwave

**We kept the original sentence**

319: decrease of the surface temperature

**We kept the original sentence**

321: why does the persistent cloud cover reduce the temperature gradient?

**Because air temperature decrease, reaching values close to the surface temperature.**

326: remove "according to…"?

**Removed and changed**

328: remove "and the accumulation and ablation"

**Removed**

329: May-March?

**April-March. Thank you**

330: wit the mass balance of 2019/20

**Added**

330: The mass balance in 2018/19 was

**Added**

330: while the mass balance in 2016/17

**Added**

331: with a positive value

**We kept original sentence**

331: maybe add "cumulative" somewhere, to make sure that you don't refer only to April values. Also, it would be better to say that is "the trend" that is interrupted, I think.

**We kept original sentence**

336: The mass balance in 2020/21 shows

**We kept original sentence**

339: earlier than

**Changed**

345: please reword, the glacier experienced the mass loss, not the years.

**Agree. Changed.**

347: a sensitivity -> a hypothetical

**Deleted**

348: delete "negatively"

**Deleted.**

352: Cumulative glacier mass balance, or why "mean"? is that a spatial average?

**It is a spatial mean. We added this information in the caption.**

352: delete all "mean"

**Previous answer.**

353: are -> indicate, represent

**Not changed**

357: Hypothetical glacier mass balance

**Added**

362: using a constant lapse rate for air temperature is not realistic.

**Added**

364: The explanation of the snow albedo parameterization should on "Methods"

**We kept here as is listed as uncertainty source and the limitation are discussed.**

364: ice albedo is assumed as spatially uniform and spatially constant.

**We add spatially constant**

365: the ice albedo parameterization

**Added**

367: of the glacier surface

**Changed**

368: … penitentes – spiky…- has been noted on the Olivares River sub-basin glaciers.

**Changed**

370: "Indeed", repetition

**Deleted**

372: context of a severe drought

**We kept mega-drought**

375: the relative impact

**Added**

377-378: What do you that no direct measurements of energy balance fluxes are available? The other studies use probably the same type of data (AWSs)

**The sentence is for Olivares Alfa glacier specifically. We mention that "…similar modelling and observational analyses have been conducted on nearby glaciers…"**

387: those from previous studies

**Added**

388: such as varying periods, elevation,

**Added**

391: Geodetic mass balance of Olivares Alfa Glacier for the last 20 years has been negative.

**We kept original sentence**

393: Can you add the uncertainty of the Hugonnet estimates? I guess it is quite a lot for such a short period

**This comparison is qualitative, just to check if our modelling is in the range of previous estimation. To keep it simple we don´t includes uncertainties, which it is possible to look in the sources.**

397: Mention somewhere that the monitoring of Echaurren is done in-situ using the glaciological method

**Added**

397: Not only is reported to WGMS, but is also the reference glacier for the Southern Andes

**Agree, but we don't add this information. There are discussions about the representatively of this glacier considering the changes in its characteristics**

406: snow accumulation on

**Changed**

408-410: Can you reword using less commas? It would help the flow of the reading

**We delete a part of this sentence**

410: The maximum IVT value? Or the mean?

**Maximum.**

412: the long duration

**Added**

413: rate of snow accumulation -> snow accumulation values compared to previous summers

**We kept the original sentence**

416: determine the occurrence of snow accumulation on glaciers

**Reworded**

435: summer snow accumulation is not unusual. To illustrate this, …

**Added**

441: "after the event", for how long?

**Added**

442: , even at the elevation of …

**Changed**

443: the glacier surface

**Added**

445: However, these conditions…

**Added**

447: GEONOR sensor

**Added**

451-453: Please check the sentence structure

**Reworded**

459: of incoming longwave radiation

**Thank you**

459: they represent an energy sink

**Added**

461-464: Please reword. It is not very clear.

**Reworded**

475-475: Same here. Please consider splitting the sentence.

**We deleted a part of this sentences as we agree was confused**

480: The mass balance would have been

**Changed**

485: Maybe delete "during 1991-2021", it confuses a bit the sentence

**Ok**

485: Delete "rate"

**Ok**

501: orographic summertime precipitation

**Deleted "heavy"**

503: delete "in the subtropical Andes"

**We kept it.**

508: has -> had

**Changed**

511-512: The reduction in the magnitude of energy sinks did not compensate for the decrease in energy inputs.

**We kept the original sentence.**

519-521: Please reword, not very clear with so many commas.

**We reworded this part adding new information.**

521: accumulations -> accumulation

**Changed.**

527-530: Check the verb tenses (all in present)

**Ok**

---

## Author Response (AR3)

**Author Response:**

**We appreciate the constructive review and comments by Editor Dr. Emily Collier. We agree with the comments and we will introduce changes in the manuscript to address the editor's concerns. We corrected the manuscript considering the technical correction. We also corrected some of the figure to be consistent with the manuscript (i.e. change "elevation" by "altitude") and also to correct the Figure 2 (in the last version there was an omission in the colour scale on panel c). Finally, we change the Table S2 (S1 in the previous version), deleting the AR between 2014 and 2021 catalogued as 0. Here, we provide a brief point-by-point response to the comments and concerns by the Editor (in bold):**

Dear Authors,

Thank you for submitting your revised manuscript and responses following the second round of review. Your manuscript was well received by the reviewers, and I am pleased to accept it for publication in The Cryosphere subject to addressing the minor comments provided below.

Best regards,

Emily Collier

Minor comments:

1. Lines 102-103 and lines 240-241: The sentences stating that "the storm was usually strong, even by winter standards" and "Valenzuela et al. (2022) also showed the extreme values of this precipitation event…" may be why Reviewer #1 asked for the all-season precipitation comparison in their major comment #2. Please indicate earlier in the introduction/methods that while the storm strength was exceptional, the precipitation amount was only exceptional for dry season (at least in your study region, located on the northern boundary of the high IVT, cf. Fig. 3a).

**Thank you for your comment. We will maintain that the amount of precipitation is exceptional. The annual precipitation in this region ranges between 200 and 700 mm, so, actually 100 mm is extraordinary. We agree this amount of precipitation are more common in winter, but is in the range considered as extreme for winter (percentile 95-100). We will clarify this with a new reference indicating that over 40 mm per day in winter is considered extreme (Garreaud, 2013), so the total precipitation of the event is close or in the range of extreme winter events.**

2. Dr. Ayala requested that the other potential contributors to the near-neutral MB (the subsequent small snowfall events and cooler temperatures) be mentioned in the abstract. Please add this information at lines 28-30. Related to phrasing of the results, at line 334 you state that "five of all seven years finished with negative mass balance" and at line 347 that "the mass balance of this hydrological year was among the least negative." However, the simulations show three years with clearly negative MB, three with near-neutral MB, and one with clearly positive MB. Please rephrase these sections along the lines of what is expressed elsewhere in the manuscript, namely that the AR results in an MB near equilibrium despite strong ablation in the preceding months.

**We introduce the changes suggested by the Editor, rephrasing the description of the mass balance results.**

3. The extension of the AR analysis using ERA5 to the period of 1941-2023 is an excellent addition to the revised manuscript and demonstrates how exceptional the studied AR was. Currently, in the SI, only information about ARs detected between April 2013 to March 2021 is provided. Please provide a supplementary csv file with all detected events over the full period.

**Thank you for the comment. We added the catalogue of the summertime AR in the supplementary material and cited in the manuscript.**

4. With regards to the glacier mass balance simulations with COSIPY, can you please add some information about how the free parameters were specified for your study region? Please also rephrase lines 202-203 to clarify that the model provides a utility (i.e. it is not an intrinsic functionality) for distributing point forcing data and that the lapse rates and grid spacing employed are user-defined. Please indicate which lapse rates you selected. Finally, I assume an hourly timestep was used when running COSIPY. I therefore suggest the following change at lines 209-211: "we re-initialized the model with a snow-free initial condition for each simulation year." However, I assume this approach is more related to model drift if run continuously from 2014-2021 than to the scarcity of snow observations. Please add a line of discussion to the manuscript if so.

**Thank you for this comment. We clarify this in the section, following Editor suggestions. We added in the manuscript the parameters.**

5. Technical corrections:

**We added all the technical corrections listed by the Editor. We comment one of them to clarify.**

- There is some repetitive text that could be reduced:

o Lines 45-46: I suggest removing "modulating annual mass balance"

**Removed.**

o Line 67: I suggest removing the sentence "In the case of ablation…" as this information is covered in the preceding and following sentences.

**Removed**.

o Lines 95-104: Valenzeula et al. (2022) is cited after every sentence. Can you rephrase to cite the study fewer times?

**Corrected.**

o Lines 130-131: I suggest rephrasing "at two different scales: first, for the Maipo River basin and second, its Olivares River sub-basin"

**Changed.**

o Lines 134: I suggest rephrasing "The basin contains around 1000 ice bodies comprising a total glacier area of 388 km2 (Barcaza et al., 2017), while the sub-basin hosts a glacierised area of…"

**Changed.**

o Lines 2323: I suggest "The climatological…" and removing "on the Pacific coastal grid points (30°S-35°S" and "(83 years)" as the reader has just seen this information in the methods.

**Changed.**

o Define "summer season (DJF)" and "atmospheric river (AR)" only once

**Corrected.**

- Line 65: please move the information about the geographic location of Brewster glacier from line 458 to here

**Moved.**

- Lines 114-116: I suggest introducing the exact model employed in the methods section and instead adding that you performed physically based modelling to the previous sentence.

**Corrected.**

- In Figure 1, the weather station at Olivares Alfa Glacier is labelled 'AWS' but referred to as 'AWSOA' in many places in the text

**Changed.**

- Please refer to the studied event in a consistent way throughout the manuscript ("2021 AR", "summertime-2021 AR", "**summer-2021 AR**", "January 2021 AR," and "summer AR" are all used)

**We used summer-2021 AR.**

- Line 253: Figure 3c?

**Figure 3a is fine as is the synoptic characteristic derived from the analysis of that panel, specifically for the event Figure 3c gives the comparison with previous summer ARs,**

- Lines 299-302: "The information about HYSPLIT in this section comes out of the blue. I suggest either introducing in the methods that you use this trajectory tool offline or saying more generally "Offline trajectory analysis indicates the discrepancy could be related to the radiosonde trajectory, as it travelled towards the Andes on 29 and 30 January but northwest over the Pacific Ocean on 2 February (see Fig. S2)."

**Changed as suggested.**